# Proteomic landscape of tunneling nanotubes reveals CD9 and CD81 tetraspanins as key regulators

Roberto Notario Manzano[1,2], Thibault Chaze[3], Eric Rubinstein[4], Esthel Penard[5], Mariette Matondo[3], Chiara Zurzolo[1*†], Christel Brou[1*†]

[1]Membrane Traffic and Pathogenesis Unit, Department of Cell Biology and Infection, CNRS 18 UMR 3691, Institut Pasteur, Université Paris Cité, Paris, France; [2]Sorbonne Université, ED394 - Physiologie, Physiopathologie et Thérapeutique, Paris, France; [3]Proteomics Platform, Mass Spectrometry for Biology Unit, CNRS USR 2000, Institut Pasteur, Paris, France; [4]Centre d'Immunologie et des Maladies Infectieuses, Inserm, CNRS, Sorbonne Université, CIMI-Paris, Paris, France; [5]Ultrastructural BioImaging Core Facility (UBI), C2RT, Institut Pasteur, Université Paris Cité, Paris, France

*For correspondence:
chiara.zurzolo@pasteur.fr (CZ);
christel.brou@pasteur.fr (CB)

†These authors contributed equally to this work

Competing interest: The authors declare that no competing interests exist.

**Abstract** Tunneling nanotubes (TNTs) are open actin- and membrane-based channels, connecting remote cells and allowing direct transfer of cellular material (e.g. vesicles, mRNAs, protein aggregates) from the cytoplasm to the cytoplasm. Although they are important especially, in pathological conditions (e.g. cancers, neurodegenerative diseases), their precise composition and their regulation were still poorly described. Here, using a biochemical approach allowing to separate TNTs from cell bodies and from extracellular vesicles and particles (EVPs), we obtained the full composition of TNTs compared to EVPs. We then focused on two major components of our proteomic data, the CD9 and CD81 tetraspanins, and further investigated their specific roles in TNT formation and function. We show that these two tetraspanins have distinct non-redundant functions: CD9 participates in stabilizing TNTs, whereas CD81 expression is required to allow the functional transfer of vesicles in the newly formed TNTs, possibly by regulating docking to or fusion with the opposing cell.

## eLife assessment

Notario Manzano et al. offer a **valuable** first analysis of proteins within tunneling nanotubes (TNTs), membranous bridges connecting cells. This work distinguishes TNTs from extracellular vesicles, but further experimental and analytical tools are needed to refine the TNT proteome. **Solid** data supports a role for tetraspanins CD9 and CD81 in TNT function. The proposed model for CD9 and CD81 is over-interpreted and requires additional evidence for stronger support.

## Introduction

Tunneling nanotubes (TNTs) are thin membranous conduits, supported by F-actin that form continuous cytoplasmic bridges between cells over distances ranging from several hundred nm up to 100 μm (*Rustom et al., 2004*; *Sartori-Rupp et al., 2019*). They allow cell-to-cell communication by facilitating the transfer of different cargoes directly from cytoplasm to cytoplasm of the connected cells, including organelles (e.g. lysosomes, mitochondria) (*Abounit et al., 2016*; *Pinto et al., 2021*; *Cordero Cervantes and Zurzolo, 2021*), micro- or mRNAs, pathogens (*Connor et al., 2015*; *Haimovich et al., 2017*) and misfolded/aggregated proteins (e.g. prion proteins, tau, or α-synuclein aggregates *Vargas et al., 2019*; *Dilsizoglu Senol et al., 2021*; *Chastagner et al., 2020*). TNTs could play major roles in

various diseases, including neurodegenerative diseases (*Tarasiuk and Scuteri, 2022*; *Victoria and Zurzolo, 2017*; *Zurzolo, 2021*) or cancers of different types (*Pinto et al., 2020*). In addition to cell cultures and tumor explants (*Pinto et al., 2020*), TNT-like connections have been shown to exist in the retina and facilitate cellular material transfer between photoreceptors (*Kalargyrou et al., 2021*; *Ortin-Martinez et al., 2021*) or pericytes (*Alarcon-Martinez et al., 2020*), highlighting the importance of understanding the biology of these protrusions in order to unravel their possible role(s) in vivo (*Victoria and Zurzolo, 2017*). TNT formation is highly dynamic and appears to be regulated by cellular stresses and actin regulators (*Ljubojevic et al., 2021*). Two models have been proposed for TNT formation. In the first, TNTs would be produced by membrane deformation and elongation of the protrusion supported by actin polymerization followed by adhesion and fusion of this protrusion with the opposing cell. Alternatively, TNTs could be formed by cell dislodgement where adhesion and membrane fusion would be the first steps before cell body separation and elongation of the protrusion by actin polymerization (*Ljubojevic et al., 2021*; *Abounit and Zurzolo, 2012*). However, TNTs have been shown to be structurally more complex, being made up of bundles of fine open connections (called iTNTs) held together by partially identified proteins, including N-Cadherin (*Sartori-Rupp et al., 2019*). The mechanism(s) and specific pathways governing iTNT/TNT formation as well as the molecular components of TNTs are still not known.

In addition to TNTs, one of the major pathways used by cells to transfer materials over long distances is through membrane-surrounded vesicles, collectively known as extracellular vesicles (EVs) (*Mathieu et al., 2021*). EVs are released by all cells, and up taken by distant recipient cells (*Charreau, 2021*; *van Niel et al., 2022*; *Wolfers et al., 2001*). They can be formed either by direct budding from the plasma membrane, or by secretion of intraluminal vesicles of multivesicular compartments (in which case they are called exosomes). Because of their common functions, their similar diameters, and because TNTs are fragile and easily broken and therefore can be released in cell culture supernatants (where EVs are also found), it has been challenging to distinguish TNTs from EVs in terms of composition (*Gousset et al., 2019*). In this regard, two members of the tetraspanin family, CD9 and CD81, which are well-known and widely used markers for EVs (*Théry et al., 2018*), have been detected in TNTs in T cells when overexpressed (*Lachambre et al., 2014*).

Tetraspanin forms a family (with 33 members in mammals) of small four membrane-spanning domain proteins with two extracellular domains, including a large one harboring a tetraspanin-specific folding and two short cytoplasmic tails. Tetraspanins are involved in various cellular processes like migration, adhesion, signaling, pathogen infection, membrane damage reparation, membrane protrusive activity, and cell-cell fusion (*Charrin et al., 2009*; *Charrin et al., 2014*; *Hemler, 2005*; *Huang et al., 2018*). Their function is linked at least in part to their ability to interact with other transmembrane proteins, forming a dynamic network or molecular interactions referred to as the tetraspanin web or Tetraspanin-enriched microdomains (TEM) (*Yáñez-Mó et al., 2009*). Inside this 'web,' the tetraspanins CD9 and CD81 directly interact with the Ig domain proteins CD9P1 (aka EWI-F, encoded by the PTGFRN gene) and EWI-2 (encoded by the IGSF8 gene) (*Stipp et al., 2001b*; *Stipp et al., 2001a*; *Charrin et al., 2001*; *Charrin et al., 2003*), which have an impact on several fusion processes (*Charrin et al., 2013*; *Cohen et al., 2022*; *Whitaker et al., 2019*). Whether CD9 and CD81 are also endogenously present in TNTs from non-T lymphoid cells, and whether they play a role in the formation or function of TNTs has not been investigated.

With the goal of identifying structural components of TNTs, and possibly specific markers and regulators of these structures, we established a protocol of TNT isolation from U2OS cultured cells (*Pontén and Saksela, 1967*), allowing to separate TNTs from extracellular vesicles and particles (EVPs) and from cell bodies. We obtained the full protein composition of TNTs, compared to EVPs. As CD9 and CD81 were major components of TNTs, we further studied their specific roles in TNT formation and ability to transfer cellular material using human neuronal SH-SY5Y, a well-known cell model to study functional TNTs (*Sartori-Rupp et al., 2019*; *Chakraborty et al., 2023*). Our data indicate that CD9 and CD81 have different and complementary functions by possibly regulating two different steps of the formation of TNTs.

## Results

### Purification of TNTs and EVPs

In order to isolate nanotubes and similar structures (generically referred to hereafter as TNTs), we took advantage of the fact that TNTs are very sensitive to mechanical stress as they are not attached to the substrate (*Rustom et al., 2004*; *Pontes et al., 2008*). We used U2OS cells because they are robustly adherent cells exhibiting few long protrusions and are able to grow TNTs in complete and serum-free medium and transfer cellular material through these bridges (*Figure 1—figure supplement 1A, B* and *Pergu et al., 2019*). Cells were first plated and cultured in serum-free medium to facilitate the purification steps; next after removal of the cell medium and replacement by a small volume of PBS, the flasks were vigorously shaken to break the TNTs that were then isolated from the supernatant by ultracentrifugation after elimination of floating cells by low-speed centrifugations and filtration (see workflow on *Figure 1A*). When necessary, EVPs were directly collected from the first cell culture supernatant and enriched following a standardized procedure (*Théry et al., 2018*; *Cocozza et al., 2020*; *Théry et al., 2006*; *Alvarez et al., 2012* and see *Figure 1A*). Observation of the obtained particles using transmission electronic microscopy revealed that they were morphologically different (*Figure 1B*). EVPs appeared mostly circular with a mean diameter of 60 nm (*Figure 1C*), whereas TNTs (or TNT fragments) were mostly cylindrical, with a mean diameter of 69 nm and length varying from 140 to 900 nm (mean length 372 nm). These results were in accordance with the expected sizes of EVPs, and with the different nature of the materials of each fraction. To further validate that our protocol for preparing TNTs vs. EVPs was accurate, we sought to engineer cells with fluorescent TNT markers. As shown in *Figure 1—figure supplement 1A*, actin chromobody-GFP (an actin-detecting probe that does not affect actin dynamics *Melak et al., 2017*) decorated TNTs after stable expression. Endogenous CD9 was present on at least a fraction of TNTs formed between U2OS cells (cultured in serum-free or -rich conditions, *Figure 1—figure supplement 1A*), and a GFP-CD9 construct also decorated these structures when stably expressed in these cells (*Figure 1—figure supplement 1B*). As a control, cells stably expressing H2B-GFP (labeling nuclei) were used.

Four independent preparations of TNTs and EVPs from the same cell cultures (*Figure 1D*), were analyzed. We confirmed that the crude preparations (before ultracentrifugation, *Figure 1A*) contained particles and measured their size and their fluorescence using Nano flow-cytometry. As shown in *Figure 1E* and *Figure 1—figure supplement 1C–E*, both EVPs and TNTs contained particles of similar mean diameter (around 60 nm, *Figure 1—figure supplement 1C, D*), ranging from 40 to more than 100 nm, in perfect accordance with TEM results. The sizes and concentrations of EVPs and TNTs were the same in parental and transfected cell lines (*Figure 1—figure supplement 1D, E*). Both fractions contained particles bearing GFP-CD9 (*Figure 1E*), confirming the presence of membrane-enclosed particles. A major difference between the two fractions was the large proportion of particles in the TNT preparation (38%) that bore the actin chromobody-GFP which was barely detected in EVPs (0.2%). Finally, H2B-GFP was present in a very minor part of the particles, and the Golgi marker GM130 was detected neither in EVPs nor in TNTs by western-blot (*Figure 1F*), showing that contaminations with cell debris or nuclei were limited, thus validating our protocol. CD9 and CD63 were detected in both TNT and EVP preparations, whereas tubulin (shown to label a fraction of TNTs in *Figure 1—figure supplement 1A*) was only detected in TNT fraction. The classical marker of EVPs Alix was mainly detected in the EVP fraction, and the ER protein calnexin was detected in neither fraction (*Figure 1—figure supplement 1F*). We checked that shaking the cells did not affect their shape and plate attachment (*Figure 1—figure supplement 1G*, 2 left pictures). In contrast, it decreased the percentage of connections between cells, identified after mild trypsinization and phase image analysis of fixed cells (right pictures and graph of *Figure 1—figure supplement 1G*) cultured in serum-free medium. To verify that culture of the cells in the serum-free medium had no consequence on the nature of the TNTs, we compared the number of connection-forming cells in both conditions (*Figure 1—figure supplement 1G*), as well as the ability of cells to transfer DiD-labeled vesicles to acceptor cells in a cell contact-dependent manner (*Figure 1—figure supplement 1H*). We observed that connections were formed in both culture conditions, and that cell-to-cell transfer was happening in a similar ratio in serum-rich and reduced conditions. Together with the expression of several markers in both conditions on these cells (*Figure 1—figure supplement 1* and *Figure 2—figure supplement 1*), these results confirmed that serum deprivation did not affect the nature of the TNTs. Altogether, these

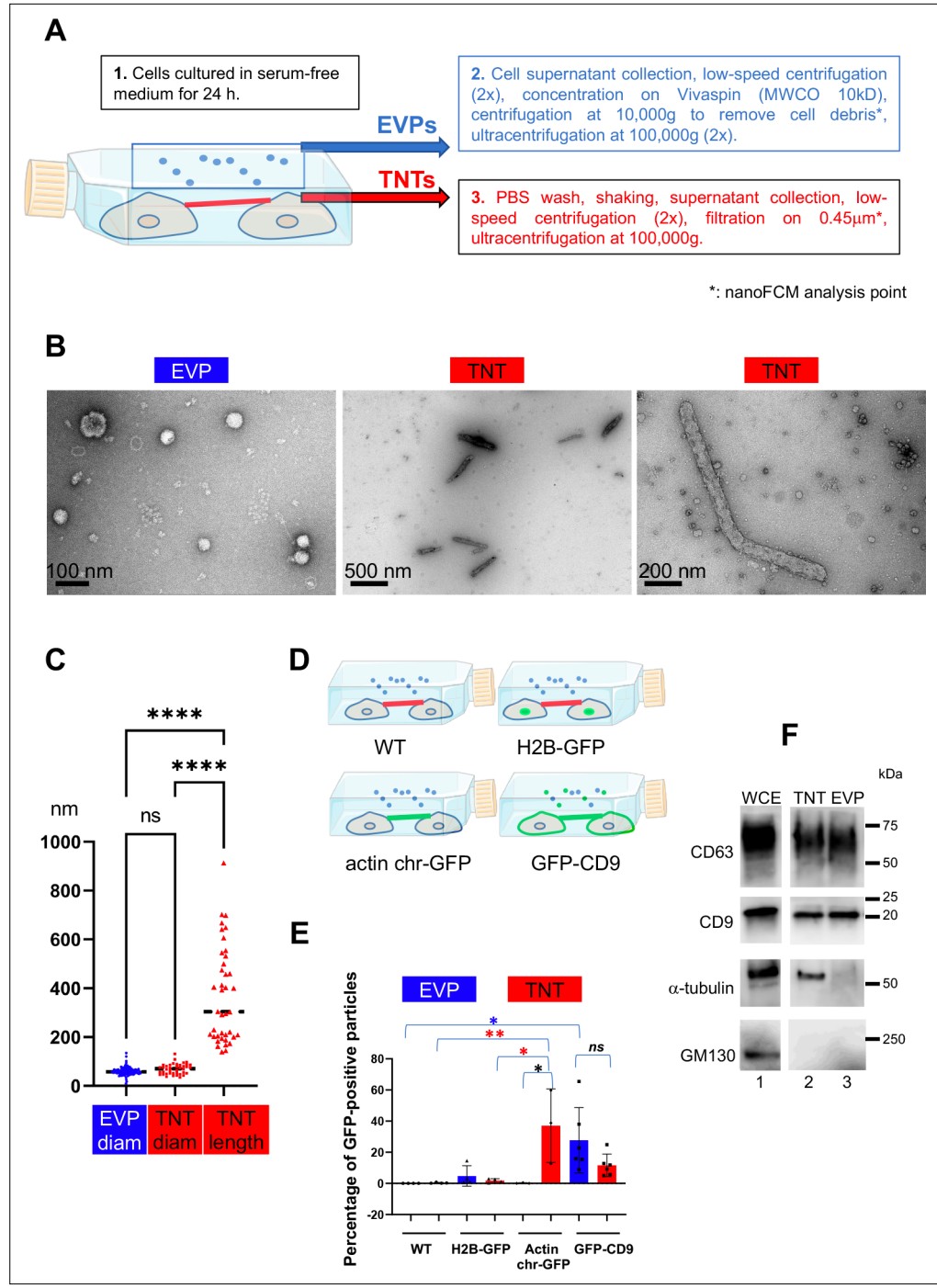

**Figure 1.** Validation of the purification procedures. (**A**) Workflow for tunneling nanotube (TNT) vs. extracellular vesicles and particle (EVP) purification. EVPs were purified from cell culture supernatant, and TNTs from the remaining attached cell supernatant after shaking. * indicates the fraction used for nano-flow cytometry (NanoFCM) analysis. (**B**) Representative pictures of negative staining and transmission electron microscopy from EVP and TNT fractions as indicated. Scale bars are 100, 500, and 200 nm, respectively. (**C**) Violin plot of the size distribution of EVP diameters (124 vesicles) and TNT diameters and lengths (40 objects), line is the median. EVP and TNT diameter means are 60 and 69 nm, respectively, TNT lengths extend from 140 to 912 nm, mean of 372 nm. Statistical analysis is one-way Anova with Tukey post hoc correction. Ns, non-significant, ****$p<0.0001$. (**D**) Schematic representation of stable cell lines where green color indicates the location of GFP-tagged protein: H2B-GFP (nuclear), actin chromobody-GFP (actin cytoskeleton including TNTs) and GFP-CD9 (cell surface, TNTs, EVPs). (**E**) Scatter dot plot representing the mean percentage (with SD) of GFP-positive particles analyzed by

*Figure 1 continued on next page*

*Figure 1 continued*

nanoFCM. Statistical analysis of three independent experiments to 6 for GFP-CD9 (oneway Anova with Tukey post-hoc correction) show the following respective p-values (from top): blue*: 0.0252, red**: 0.0089, red*: 0.0131, black*: 0.0161. Means values are (from left to right): 0.01, 0.3, 4.8, 1.75, 0.16, 37.07, 27.7 and 11.6. (**F**) Western blot of whole cell extracts (WCE) (20 μg, corresponding to around $0.1 \times 10^6$ cells), TNT, and EVP fractions (both from 10 $10^6$ cells) prepared from the same cells, blotted with CD63, CD9, α-tubulin and GM130 specific antibodies. White lane indicates that intervening lanes of the same gel (and same exposure) have been spliced out.

The online version of this article includes the following source data and figure supplement(s) for figure 1:

**Source data 1.** Uncropped and labeled western blots (WBs) for *Figure 1F*.

**Source data 2.** Raw unedited western blots (WBs) for *Figure 1F*.

**Figure supplement 1.** Characterization of tunneling nanotubes (TNTs) in U2OS cells.

**Figure supplement 1—source data 1.** Uncropped and labeled western blots (WBs) for *Figure 1—figure supplement 1F*.

**Figure supplement 1—source data 2.** Raw unedited western blots (WBs) for *Figure 1—figure supplement 1F*.

data supported that our protocol allowed differential enrichment of the fractions in TNTs and EVPs respectively.

## Analysis of TNTome

To obtain a full and accurate picture of U2OS TNT content, we made 12 independent preparations of TNT fractions (in red in *Figure 1A*), each starting from about $20 \times 10^6$ cells, and analyzed them by LC-MS/MS. These fractions are enriched in the TNT-microsomal-type/membrane proteome, although we cannot exclude that they also contain small portions of cell bodies or ER. For simplicity, we shortly call them TNTome hereafter. 1177 proteins were identified in at least nine preparations (*Supplementary file 1*). We first observed that proteins previously described in TNTs, like actin, Myosin10 (*Gousset et al., 2013*), ERp29 (*Pergu et al., 2019*), or N-cadherin (*Sartori-Rupp et al., 2019*; *Chang et al., 2022*) were indeed present in the TNTome. Less than 100 nuclear proteins (according to GO Cellular component analysis), i.e., less than 8%, were found, which could result from partial contamination with cellular debris or dead cells. This is in accordance with NanoFCM results, where H2B-GFP positive particles were 4% of actin-chromobody-GFP positive ones. The 1177 proteins have been ranked in four quartiles depending on their relative abundancy (average iBAQ), highlighting the enrichment of specific factors when considering gene ontology (*Supplementary file 1*, *Figure 2A*). These factors could be structural components of TNTs, but also material that was circulating in them or in the process of being transferred at the time when TNTs were broken and purified. It could be why mitochondrial (8% of the total), lysosomal/endosomal or other vesicle proteins are listed. The Proteomap analysis (*Figure 2A*) also revealed that TNTome is rich in RNA-associated proteins (ribosomes, translation factors, ribonucleoproteins), in accordance with TNTs being able to transfer micro and mRNAs (*Haimovich et al., 2021*; *Kolba et al., 2019*).

A major group of proteins of interest were related to the cytoskeleton (15%, i.e. 172 proteins, see *Supplementary file 2*, analysis of GO terms, cellular components). As shown by STRING functional network representation, actin-related proteins were majority (*Figure 2B*, orange nodes) compared to microtubule-related (green nodes) or intermediate filament-related proteins (blue nodes). Actin was the fourth most abundant protein of TNTome, whereas tubulin beta and alpha chains were ranked in positions 6 and 7. This was in accordance with the nature of TNTs, mainly supported by actin cytoskeleton (*Rustom et al., 2004*). However, recent work has shown that TNTs, in addition to actin, could contain microtubules as in some cancers (*Figure 1—figure supplement 1A* and *Pinto et al., 2020*; *Lou, 2020*), and cytokeratins like in the case of urothelial cells (*Resnik et al., 2019*; *Resnik et al., 2018*).

When looking at membrane proteins, analysis of the GO terms (cellular components) of the TNTome classified around 500 proteins as membrane-related, 64 of which being strictly integral plasma membranes (see *Supplementary file 3* and *Figure 2C*). Among the latter, N-cadherin and other cadherin-related proteins (green nodes), as well as known N-cadherin interactors like alpha-catenin, were found. We also noticed the presence of various integrin subunits (orange nodes), including the Integrin β1. However, the TNTome had only a partial overlap with integrin adhesion

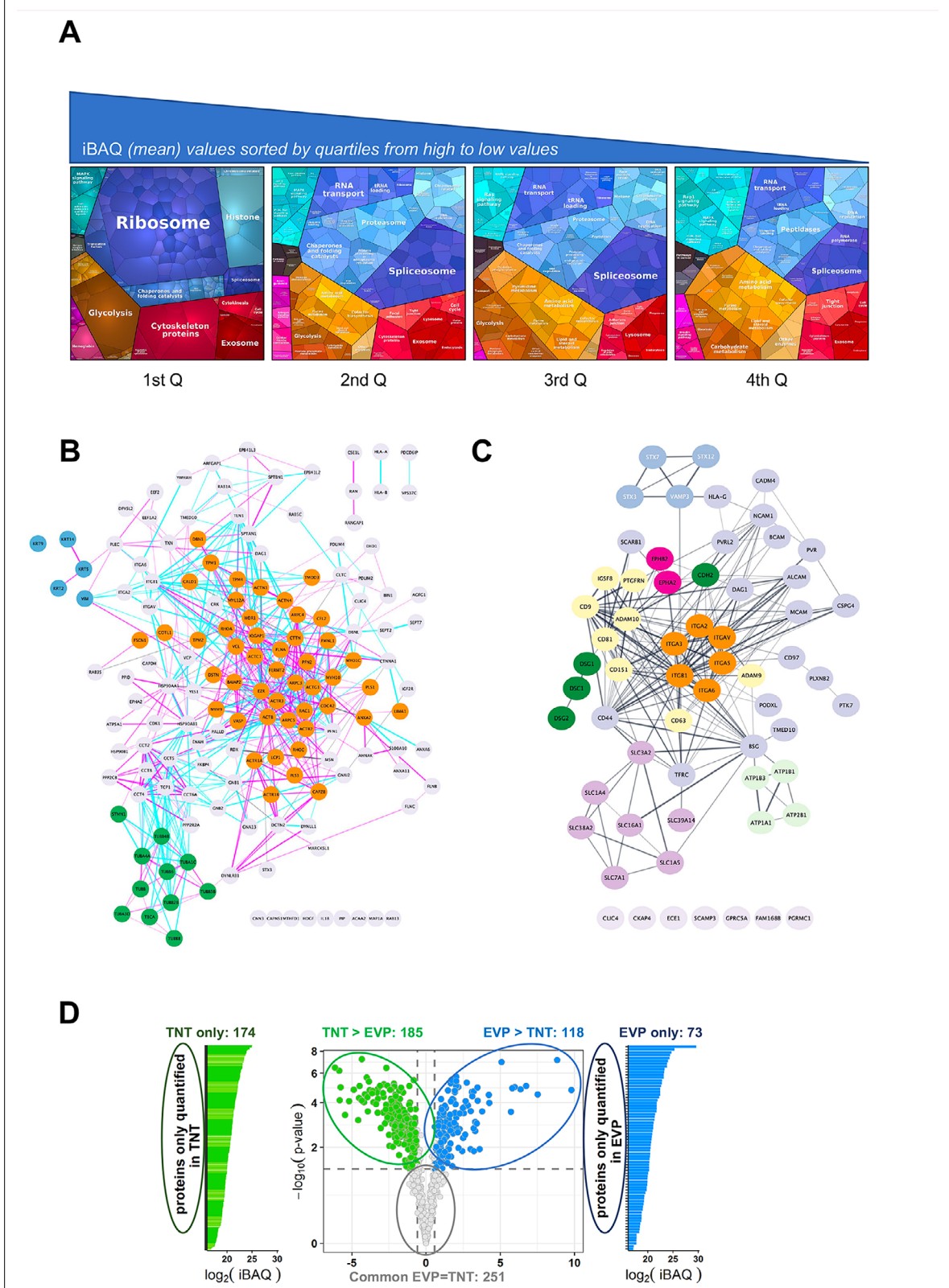

**Figure 2.** Analysis of the TNTome. (**A**) Proteomap of the 1177 proteins of the TNTome, sorted in four quartiles depending on their mean iBAQ. Protein accession and mean iBAQ were used to create ProteoMap, analyzed according to Gene Ontology. (**B**) STRING physical association network for the cytoskeleton-related proteins listed in *Supplementary file 2*. Color groups were created using Cytoscape. Green are microtubule-related proteins, blue are intermediate filaments, and orange are actin-interacting proteins. Blue and pink edges show physical interactions based on databases and

*Figure 2 continued on next page*

*Figure 2 continued*

experiments respectively. (**C**) Full STRING functional association network for integral surface membrane proteins of TNTome, based on ***Supplementary file 3***. Orange are Integrin proteins, red are Ephrin receptors, dark green are Cadherins, light green are Sodium/potassium transporting ATPase ions channels, purple are monocarboxylate and amino acids transporters, and yellow are tetraspanin-related proteins. (**D**) Volcano plot of the mass spectrometry analysis based on the four extracellular vesicles and particle (EVP) and tunneling nanotube (TNT) preparations, showing the maximum log2(Fold-change) in the x-axis measured between TNT and EVP fractions and the corresponding -log10 (p-value) in the y-axis. Dashed lines indicate differential analysis quadrants with log2 (Fold-change)=0.58 and false discovery rate FDR = 1%. Common EVP = TNT is non-significantly different (FDR >0.05) with FC >1.5, and FC <1.5. Each quadrant is named above and the number of identified proteins is indicated. Left and right are proteins non-overlapping in both fractions: TNT-only and EVP-only. Note that in EVP-only fraction, 10 proteins were found in TNTome (based on 12 experiments) and should therefore be removed. For the TNT proteins, only the proteins also present in the TNTome have been counted.

The online version of this article includes the following source data and figure supplement(s) for figure 2:

**Figure supplement 1.** Comparison of TNTome with Integrin adhesome and other cell proteins.

**Figure supplement 1—source data 1.** Uncropped and labeled western blots (WBs) for *Figure 2—figure supplement 1G*.

**Figure supplement 1—source data 2.** Raw unedited western blots (WBs) for *Figure 2—figure supplement 1G*.

**Figure supplement 1—source data 3.** Uncropped and labeled western blots (WBs) for *Figure 2—figure supplement 1H*.

**Figure supplement 1—source data 4.** Raw unedited western blots (WBs) for *Figure 2—figure supplement 1H*.

complexes (*Horton et al., 2015*, *Figure 2—figure supplement 1A* and tab1 of *Supplementary file 4*) and with the consensus adhesome (*Figure 2—figure supplement 1B*, tab2 of *Supplementary file 4*). In addition, very important proteins of the focal adhesions, like Paxillin, GIT2, Parvins, and PINCH1 were not in the TNTome, making unlikely focal adhesion being isolated in TNT preparations. We also analyzed whether TNT preparations could be contaminated by filopodia, another type of protrusion grown by the cells. Despite common proteins described as core filopodia proteins in U2OS cells (like Myosin10, Integrin α5, TLN1, FERMT2, MSN, LIMA1), the TNTome was devoid of others (Myo15A, TLN2, PARVA, ITGB1BP1, see *Jacquemet et al., 2019*), ruling out an important contamination of TNTs with filopodia during the preparation. Together, these results suggested that the proteins identified in TNTome represent a specific composition of these protrusions, and not a pool of other types of protrusions which could have contaminated the preparation. We confirmed these results by immunofluorescence both for proteins found (Integrin β1, CD151, Vinculin) or not (Paxillin, GM130) in the TNTome in U2OS cells cultured in the presence and absence of serum (*Figure 2—figure supplement 1C, D*).

We also looked at whether the TM proteins of the TNTome were the most abundant TM proteins of U2OS cells. We compared the rank in TNTome to the protein concentration (*Beck et al., 2011*) or to the level of RNA (*Lundberg et al., 2010*). Whether some factors are indeed highly expressed in U2OS cells (for instance Integrin β1, SLC3A2, basigin, CLIC channels) and ranked in the first quartile of TNTome, others are weakly expressed in U2OS cells, but still enriched in TNTome (N-cadherin for instance). In addition, some cell surface proteins are highly expressed in U2OS cells (according to their corresponding RNA level *Lundberg et al., 2010*), but not detected in the TNTome, like SLC2A11 (Glucose transporter), APP, ITM2C, TNFR. As shown by WB in *Figure 2—figure supplement 1G*, we could detect membrane or membrane-associated proteins in U2OS cell extracts that were not listed in TNTome, like Integrin β4, α4, EGFR, Cx43, and APP (*Jouannet et al., 2016*). Likewise, some proteins of very low abundancy in cells were found in TNTome (CALML5 for example), maybe reflecting a specific role in TNT formation. We also observed that the relative abundance of some proteins in TNT fractions compared to WCE (*Figure 2—figure supplement 1H*) was variable. For example, CD9 and especially ADAM10 seemed to be relatively abundant in TNTs compared to WCE, whereas Integrin β1 or ANXA2 TNT/WCE ratios were much lower. Altogether, these data suggested that TNT membrane composition was not just a fraction of cell surface membrane proteins, but rather that some proteins were excluded, other more present. This first consolidated our TNT purification procedure, and second highlighted that specific mechanisms and factors should be at stake to grow and maintain TNTs.

## Comparison of the content of TNTs and EVPs

Because both EVs and TNTs have similar characteristics (membrane-formed, diameter, ability to transfer material to remote cells), we analyzed their respective composition when prepared from the same cell

cultures, following the full protocol schematized in *Figure 1A*, from four independent experiments. 961 proteins in total were identified at least in 3 of the 4 TNT and EVP preparations. When keeping TNT proteins that were also present in the 1177 list of TNTome (in nine TNT preparations over the total of 12), a total of 801 proteins were finally differentially analyzed. Our results showed a different composition of TNTs and EVPs, although common factors represent 75% of them (see volcano plot in *Figure 2D*). Interestingly, 174 proteins were specific for TNTs when compared to EVPs (see tab1 of *Supplementary file 5*). Among the most abundant was the ER chaperone ERp29, previously shown to be required for the formation of TNTs (*Pergu et al., 2019*). When discarding organelle-associated or translation-linked proteins, 89 proteins remained in this TNT-only list (*Supplementary file 5*, tab2, constitutive). 20% of them were involved in cytoskeleton, including the positive regulator of TNTs Myosin10 (*Gousset et al., 2013*). When analyzing the proteins differentially abundant in TNT vs. EVPs (*Figure 2D* and *Supplementary file 6*), we noticed the enrichment of the cytoskeleton-related proteins, especially actin, in TNTs compared to EVPs.

The tetraspanins CD9, CD81, and CD63, classical markers of EVs were also detected in TNTs, CD9, and CD81 being among the most abundant transmembrane proteins of TNTome (*Supplementary file 3*). While CD9 and CD63 were more abundant in EVs than in TNT, CD81 was present at a similar level in both preparations. Their direct interacting partners CD9P1 and EWI2 (respectively PTGFRN and IGSF8 in *Figure 2C*) were present in TNTome, but more enriched in EVPs than TNTs. Consistent with the presence of the integrins α3β1 and α6β1, the tetraspanin CD151 which directly associates with these integrins was also detected in TNTome (*Boucheix and Rubinstein, 2001*). Finally, the presence of CD9, CD81, and CD151 in TNTs was confirmed by immunofluorescence in U2OS (*Figure 1—figure supplement 1A* and *Figure 2—figure supplement 1C*).

## Differential regulation of TNT number by CD9 and CD81

To study the role of CD9 and CD81 in the formation of TNTs, we decided to use a cellular model in which TNT structure and function have already been largely investigated, SH-SY5Y cells. These human neuronal cells form many TNTs in a complete medium (about 30% of WT cells are connected by TNTs), which, contrary to U2OS cell TNTs, can be easily quantified, and distinguished from other protrusions by fluorescent imaging (*Sartori-Rupp et al., 2019*; *Chastagner et al., 2020*; *Pepe et al., 2022*). Similar to what we observed in U2OS cells, and in accordance with the composition of the TNTome, CD9, CD81, CD151, Integrin β1 and vinculin were detected on protrusions connecting SH-SY5Y cells (*Figure 2—figure supplement 1C, E*, *Figure 3A*), whereas TNTs appeared mostly free of Paxillin, and of the Golgi marker GM130 (*Figure 2—figure supplement 1E*), as it was the case for U2OS cells. Importantly, all protrusions connecting two cells, containing actin and not attached to the substrate, which correspond to the TNT morphological definition (*Rustom et al., 2004*), bore both CD9 and CD81, indicating that these tetraspanins are good markers for TNTs (*Figure 3A*).

We then knocked-out (KO) CD9 and/or CD81 by infecting SH-SY5Y cells with lentiviral CRISPR vectors targeting the corresponding genes. Western-blot and Immunofluorescence analysis confirmed the lack of CD9 and/or CD81 expression in these cells (*Figure 3—figure supplement 1A, B*). CD9 KO cells, but not CD81 KO cells, showed a significant reduction in the percentage of TNT-connected cells compared to WT cells (*Figure 3B, C*). The % of TNT-connected cells was even lower in the double KO cells (named CD9 and CD81 KO hereafter). The role of CD9 in TNT formation and/or stabilization was confirmed by the finding that CD9 stable overexpression (OE) resulted in a significant increase in the percentage of TNT-connected cells (*Figure 3D, E*). Consistent with KO result, CD81 stable OE did not change the % of TNT-connected cells. Together these results indicated that CD9 plays a role in the formation or maintenance of TNTs and that CD81 might partially complement CD9 absence.

## Positive regulation of TNT function by CD9 and CD81 in donor cells

The next and complementary step was to evaluate the possible influence of CD9 and CD81 on the functionality of TNTs. TNT functionality is understood as the intrinsic capacity to allow the transfer of different types of cellular material through the open channel formed between the cytoplasm of different cells. This was monitored by quantifying by flow cytometry the transfer of labeled vesicles between two different cell populations ('donors' for the cells where vesicles were first labeled and 'acceptors' for the cells that received the vesicles *Abounit et al., 2015*). A similar gating strategy was applied to all experiments (*Figure 3—figure supplement 2*), and the vesicle transfer through any

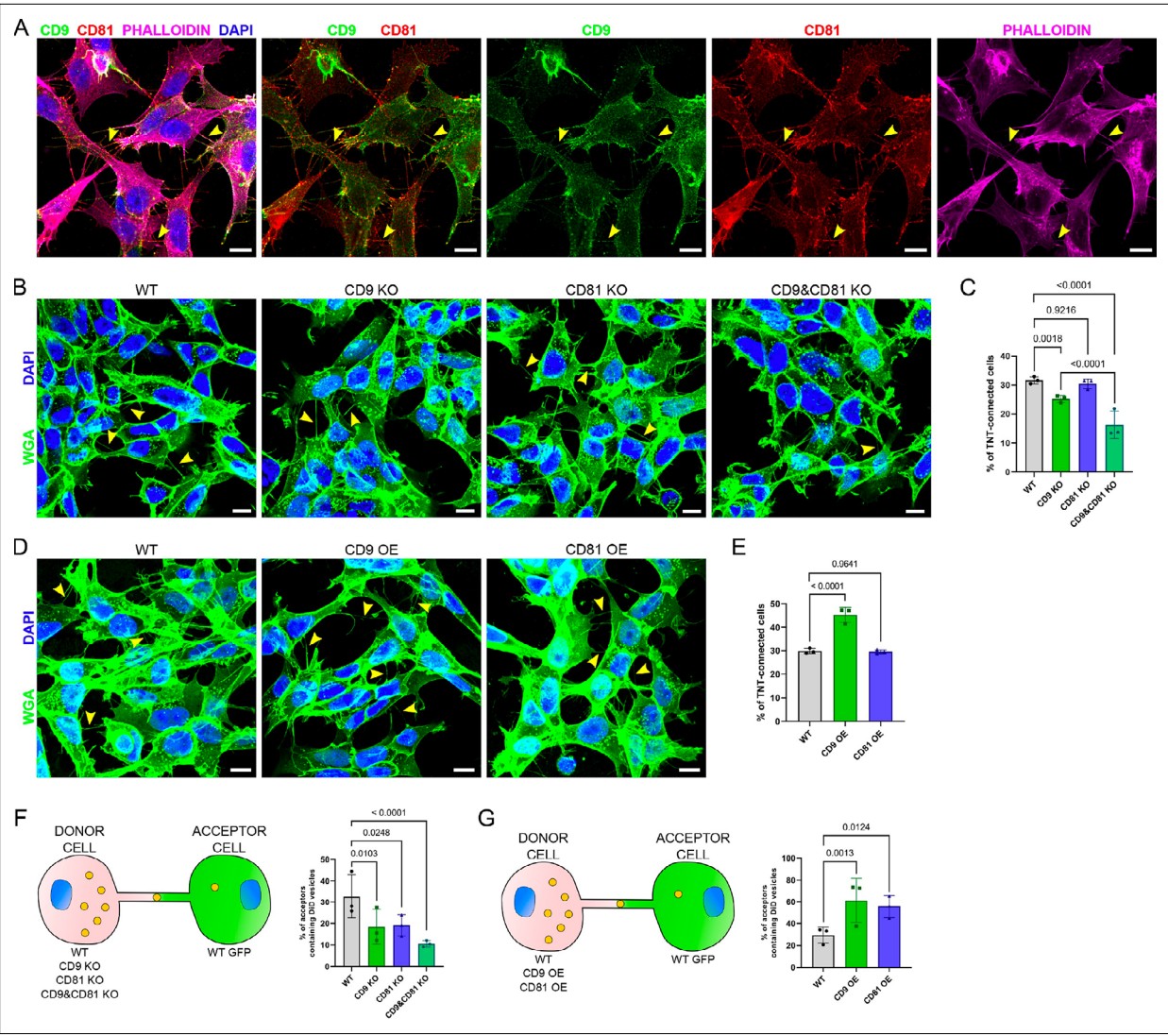

**Figure 3.** Expression of CD9/CD81 in tunneling nanotubes (TNTs) and effects of their overexpression or invalidation. (**A**) Immunofluorescence of CD9 (green) and CD81 (red) in SH-SY5Y cells. Cells were also stained with phalloidin (magenta) and DAPI (blue) to visualize actin and nuclei. This representative image corresponds to the fourth to fifth slices of a stack comprising 11 slices (1 being at the bottom). In A, B, and D, yellow arrowheads point to TNTs that are visible in the shown pictures, however, more TNTs were counted over the whole stack, which were not annotated on the figure. Scale bars correspond to 10 µm. (**B**) Representative images of TNT-connected cells in wild-type (WT), CD9 KO, CD81 KO, and CD9 and CD81 KO cells (max projection of 4–7 upper slices, z-step 0.4 µm), stained with WGA-488 (green) to label the membrane and DAPI (blue) to label the nuclei. (**C**) Graph of the percentage of TNT-connected cells in the different cells, from three independent experiments. Mean and standard deviation (SD) are: WT = 31.6 ± 1.25; CD9 KO = 25.2 ± 1.24; CD81 KO = 30.4 ± 1.65; CD9 and CD81 KO = 16.3 ± 4.81. p-values are indicated above the brackets in all graphs. (**D**) Representative images (max projection of 5–7 upper slices, z-step 0.4 µm) of TNT-connected cells in WT, CD9 OE, and CD81 OE cells, stained as in B. (**E**) Graph of the percentage of TNT-connected cells in the indicated cells from three independent experiments. Mean ± SD are: WT = 29.8 ± 1.11; CD9 OE = 45.3 ± 3.17; CD81 OE = 29.5 ± 0.84. (**F**) Vesicle transfer assay from donor cells knock-out (KO) of CD9 and CD81, as schematized on the left. Graphs are percentage of acceptor cells containing DiD vesicles from three independent experiments. Mean ± SD are: WT = 32.7 ± 10.25; CD9 KO = 18.4 ± 8; CD81 KO = 19.1 ± 4.86; CD9 and CD81 KO = 10.5 ± 1.52. (**G**) Vesicle transfer assay from donor cells OE CD9 or CD81, as schematized on the left. Graphs are percentage of acceptor cells containing DiD vesicles from three independent experiments. Mean ± SD are: WT = 29.3 ± 7.45; CD9 OE = 61.3 ± 20.44; CD81 OE = 55.6 ± 10.31.

The online version of this article includes the following source data and figure supplement(s) for figure 3:

**Figure supplement 1.** Characterization of cells knock-out (KO) and overexpression (OE) for CD9 or CD81.

**Figure supplement 1—source data 1.** Uncropped and labeled western blots (WBs) for *Figure 3—figure supplement 1A*.

**Figure supplement 1—source data 2.** Raw unedited western blots (WBs) for *Figure 3—figure supplement 1A*.

**Figure supplement 1—source data 3.** Uncropped and labeled western blots (WBs) for *Figure 3—figure supplement 1C*.

*Figure 3 continued on next page*

*Figure 3 continued*

**Figure supplement 1—source data 4.** Raw unedited western blots (WBs) for *Figure 3—figure supplement 1C*.

**Figure supplement 2.** Gate strategy and representative results of all the cocultures.

**Figure supplement 3.** Measurement of transfer by secretion in all the coculture experiments.

**Figure supplement 4.** Coculture of DiD-treated wild-type (WT), CD9KD or CD81KD donor cells with GFP-expressing cells as acceptor, and analysis of transfer by microscopy.

mechanism other than cell contact-dependent was ruled out in all experiments by analyzing secretion controls, where the two cell populations were cultured separately (see total and secretion transfers in *Figure 3—figure supplement 3*). Therefore, the vesicle transfer occurring mainly through cell-contact-dependent mechanisms is an indirect way of monitoring TNT functionality that must be analyzed with regard to TNT apparent number.

First, we co-cultured WT, CD9 KO, CD81 KO, or CD9 and CD81 KO donor cells versus WT acceptor cells expressing GFP (as schematized in *Figure 3F*). Consistently with the decrease of the % of cells connected by TNTs, CD9 KO cells showed a significantly decreased percentage of acceptor cells containing donor's vesicles compared to WT cells (*Figure 3F* and *Figure 3—figure supplement 2A*). On the other hand, despite having no effect on the number of TNTs, CD81 KO resulted in a significant reduction in vesicle transfer to acceptor cells. CD9 and CD81 KO cells showed a further decrease in vesicle transfer, consistent with the high decrease in the percentage of TNT-connected cells. We obtained very similar results when analyzing the experiment using fluorescence microscopy as a complementary approach to FACS (*Figure 3—figure supplement 4*).

Next, we performed a co-culture using tetraspanin OE cells as donor cells (*Figure 3G*). Consistent with the results of KO cells, CD9 OE significantly increased both the number of TNTs and vesicle transfer compared to WT cells (*Figure 3G* and *Figure 3—figure supplement 2B*) whereas the modest CD81 OE (*Figure 3—figure supplement 1C*) stimulated vesicle transfer without significant effect on the number of TNTs (*Figure 3E and G* and *Figure 3—figure supplement 2B*). However, given that the modest OE of CD81 was associated with a decrease in CD9 expression (albeit not statistically significant), an antagonist effect of CD81 on TNTs number would not be detectable using this model.

These data showed that both CD9 and CD81 in donor cells positively regulate the transfer of vesicles through TNTs, which could be in the case of CD9, but not CD81, a direct consequence of an increased number of TNT.

## Pathway of CD9 and CD81 in the regulation of TNTs

To further understand whether CD9 and CD81 have redundant roles or are complementary in the pathway of TNT formation, we knocked-out CD81 in cells over-expressing CD9 (quantification of the total amount of CD9/CD81 in these cells by WB in *Figure 3—figure supplement 1C*). This did not prevent the increase in the percentage of TNT-connected cells observed upon CD9 OE (*Figure 4A and B*), suggesting that CD81 does not play a role in TNT formation when CD9 is present. In contrast to WT cells (*Figure 3F*), the transfer of DiD-labeled vesicles from CD9 OE cells was no longer sensitive to CD81 KO (*Figure 4C* and *Figure 3—figure supplement 2C*), suggesting that CD9 OE can compensate for the absence of CD81 on vesicle transfer. To determine whether CD81 can compensate for CD9 function in TNT, we knocked-out CD9 in CD81 OE cells (*Figure 3—figure supplement 1C*). As shown in *Figure 4B*, CD9 KO reduced TNT number and vesicle transfer to the same extent in these CD81 OE cells (*Figure 4C* and *Figure 3—figure supplement 2C*) as in WT cells (*Figure 3C*). Thus, CD81 may not compensate for CD9 for the formation/stabilization of TNTs. This is in accordance with the two tetraspanins having different roles in the process of formation of TNTs.

## Stabilization of TNTs by CD9 AB

Knowing that the molecular conformation of CD9 molecules can induce membrane curvature (*Bari et al., 2011*; *Umeda et al., 2020*) and that induced clustering of CD9 with specific antibodies (*Nydegger et al., 2006*; *Khurana et al., 2007*) could lead to the formation of protrusions such as microvilli (*Singethan et al., 2008*), we postulated that CD9 could be involved in the initial steps of TNT formation. Consequently, we addressed whether promoting CD9 clustering could affect TNT number and function. Incubation for 2–3 hr of living WT SH-SY5Y cells with an anti-CD9 monoclonal

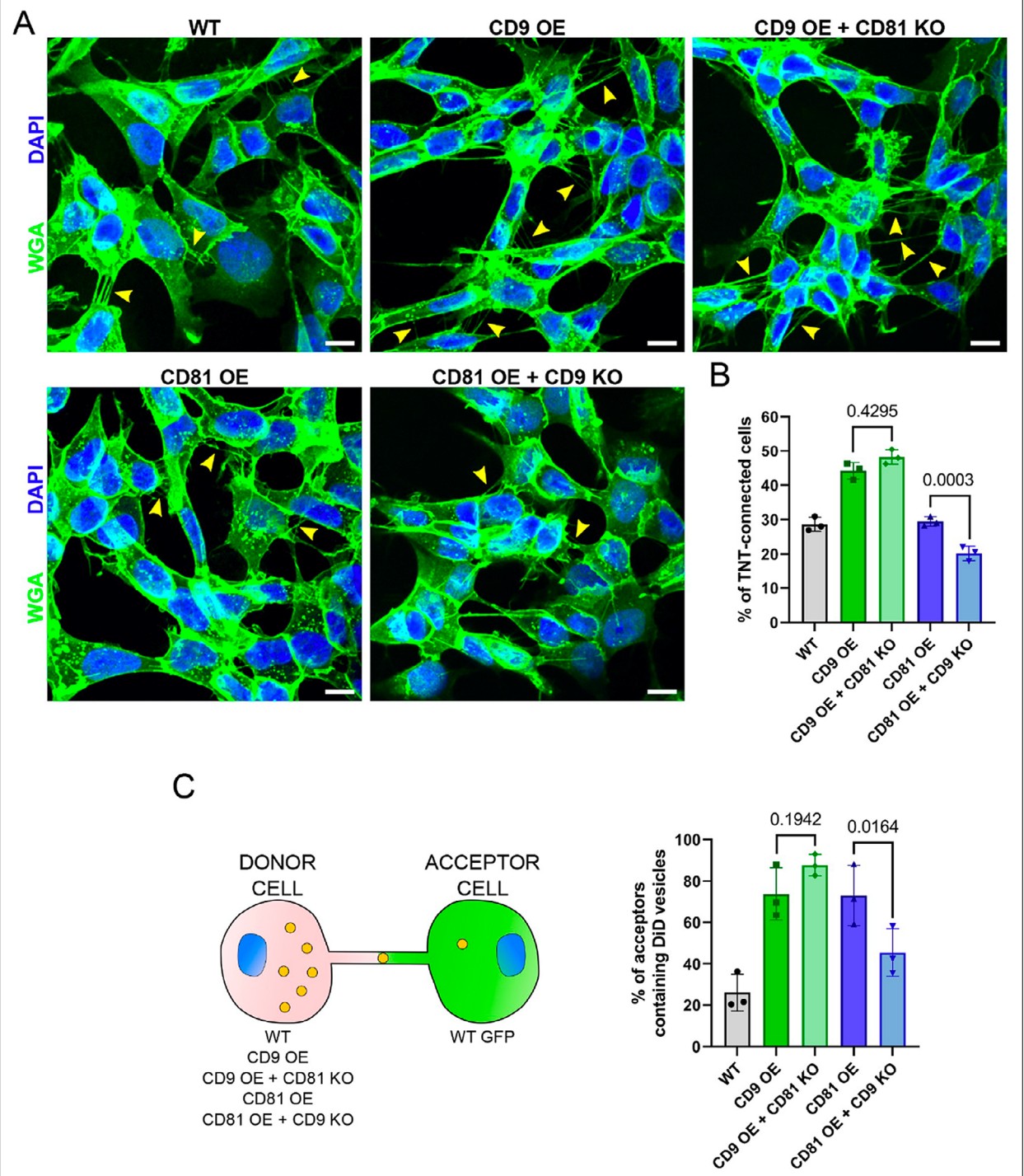

**Figure 4.** CD9 and CD81 act successively in the formation of tunneling nanotubes (TNTs). (**A**) Representative images of wild-type (WT), CD9 OE, CD9 OE + CD81 KO, CD81 OE, and CD81 OE + CD9 knock-out (KO) cells, stained with wheat germ agglutinin (WGA)-488 (green) and DAPI (blue). Yellow arrowheads show TNTs. Scale bars correspond to 10 μm. (**B**) Graph of the percentage of TNT-connected cells in the tetraspanin overexpression (OE) + KO cells. Mean ± SD (N=3) are: WT = 28.6 ± 2.05; CD9 OE = 44.2 ± 2.4; CD9 OE + CD81 KO=48.2 ± 2.16; CD81 OE = 29.5 ± 1.36; CD81 OE + CD9 KO=20.2 ± 2.09. p-values are above the brackets. (**C**) Coculture between tetraspanin OE + KO cells used as donors and WT GFP cells used as acceptors, as schematized on the left. The graph shows the percentage of acceptor cells containing DiD vesicles. Mean ± SD from three independent experiments: WT = 26.1 ± 8.94; CD9 OE = 73.8 ± 12.64; CD9 OE + CD81 KO=87.7 ± 5.11; CD81 OE = 73 ± 14.60; CD81 OE + CD9 KO=45.4 ± 11.53.

antibody (CD9 AB), but not an antibody targeting another cell surface protein (CD46 AB) or a non-specific control antibody (CTR AB), caused the relocalization of CD9 and CD81 in patches on the plasma membrane, suggesting that these molecules were incorporated in multimolecular complexes that were clustered together by the anti-CD9 antibodies (*Figure 5A* and *Figure 5—figure supplement 1*). Furthermore, CD9 AB treatment led to an increase of more than 30% in the percentage of TNT-connected cells (*Figure 5B* and *Figure 5—figure supplement 1A*). As a control, neither the CD46 AB nor the CTR AB, changed the % of TNT-connected cells (*Figure 5A and B* and *Figure 5—figure supplement 1A*). The result was similar to whether the antibodies used on living cells were directly coupled to a fluorophore (*Figure 5A and B*) or not (*Figure 5—figure supplement 1A*), thus eliminating any post-fixation artifact. Addition of CD9 AB after 24 hr coculture of WT donor and acceptor cells for an additional 2 hr resulted in an increase of the vesicle transfer of 30% (*Figure 5C*). This increase is highly significant considering that it results only from the additional 2 hour treatment over a 24 hour culture compared to control conditions. These data suggested that CD9-enriched sites could serve as initiation platforms for TNTs or participate in the stability of these structures, resulting in increased functionality. The lack of CD81 (using CD81 KO cells) did not impact CD9 clustering (*Figure 5D* and *Figure 5—figure supplement 1B*) or the increase of TNT number induced by the CD9 AB (*Figure 5E* and *Figure 5—figure supplement 1B*). However, it prevented the increase in vesicle transfer (*Figure 5F*) stimulated by the CD9 AB. This further suggests that CD9 does not require CD81 to form/stabilize TNTs and that CD81 rather regulates TNT functionality.

To determine whether the CD9 AB stabilized TNTs, we monitored TNT duration using time-lapse imaging (as previously set up *Vargas et al., 2019*) of WT or CD81KO SH-SY5Y cells stably expressing actin-chromobody-GFP. As exemplified in *Figure 6—videos 1–6* and shown in *Figure 6A–G*, CD9 AB treatment, contrary to CD46 AB or control conditions (without antibody), significantly increased the duration of TNTs from around 11 min to more than 22 min on average, both in WT and CD81KO cells (*Figure 6G*). This stabilization was accompanied by an enrichment of CD9 at the base of TNTs over time (*Figure 6B and E*), as observed on fixed cells (*Figure 5* and *Figure 5—figure supplement 1*), whereas the CD46 AB was internalized and not specifically enriched on TNTs (*Figure 6C and F*). These results are in accordance with CD9 having a role in stabilizing TNTs, CD81 not being necessary for this step.

## Regulation of TNT completion by CD81

We next tried to identify the mechanism leading to the reduced vesicle transfer from CD81 KO cells. One possibility is that vesicles cannot enter these connections in the absence of CD81. Timelapse microscopy was used to monitor the impact of CD81 KO on the behavior of DiD-positive vesicles in TNTs. *Figure 7—video 1* shows an example of a vesicle entering TNTs from WT cells and reaching the neighboring cell with constant speed and direction. *Figure 7—videos 2 and 3* show examples of vesicles from CD81 KO cells getting stuck inside TNTs or at the basis of TNTs. We quantified the content of TNTs in DiD-positive vesicles by fixing the cells 8 or 24 hr after cell labeling with DiD. *Figure 7A* shows that the proportion of TNTs containing DiD-labelled vesicles (examples in the left-hand panels) was significantly increased in CD81 KO cells compared to WT cells. Thus, the decrease in transfer observed in the absence of CD81 (*Figure 3F*) is not due to a decrease in vesicle entry into TNTs, but rather to the fact that vesicles enter TNTs but are unable to move the full length and, consequently, accumulate to some extent in the TNTs. Two hypotheses could explain this result: (i) a decreased fusion of TNT with acceptor cells or (ii) an incomplete formation of individual iTNTs in the TNT bundle (iTNTs that would not reach the opposite cell). The latter hypothesis takes into account the fact that the TNTs of SH-SY5Y cells are formed by closely apposed bundles of iTNTs which could originate from opposite cells (*Sartori-Rupp et al., 2019*) and are detected as one single TNT by confocal microscopy. Thus, CD81 KO would have an impact on TNT functionality, but not on its detection as a whole structure by confocal microscopy if TNT is formed by iTNTs originating from the two cells. To address this hypothesis, we performed cocultures (*Figure 7B*) of cells expressing Actin chromobody-GFP (considered as donor cells) with WT cells (acceptor cells). After fixation, actin and cell membranes were stained with phalloidin and WGA respectively. TNTs were identified as phalloidin (which labels all TNTs and iTNTs, no matter the cell of origin) and WGA-positive structures, whereas green fluorescence only labeled actin of iTNTs growing from donor cells, as schematized in *Figure 7B*. After max projection of the z slices encompassing the TNTs of interest, we classified the TNTs into three categories based on their

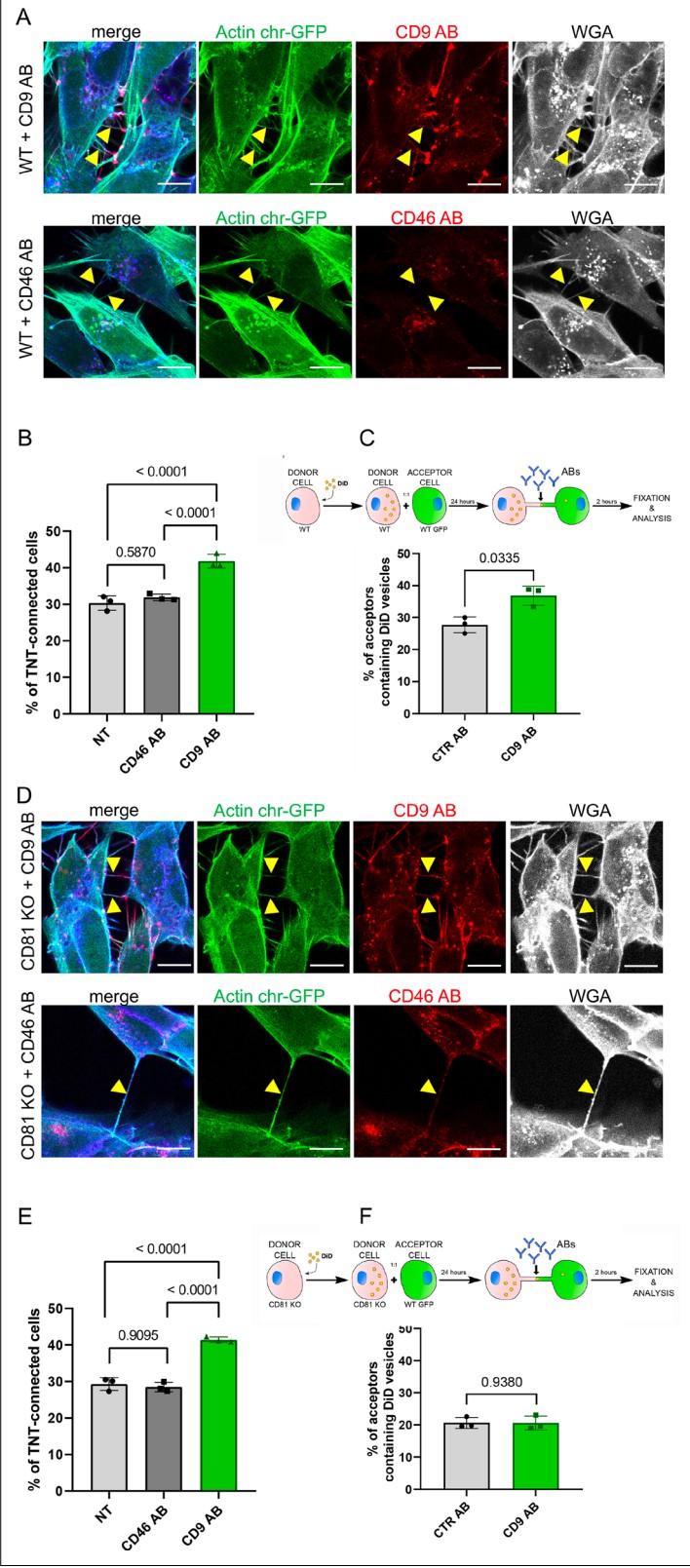

**Figure 5.** CD9 and CD46 antibody treatment in wild-type (WT) and CD81 knock-out (KO) cells. (**A**) Representative confocal images of Actin chromo body-GFP expressing WT cells, treated for 3 hr with either anti-CD9 antibody coupled to Alexa Fluor 568 (upper row), or with anti-CD46 antibody coupled to to Alexa Fluor 594 (second row). Alexa-647 coupled wheat germ agglutinin (WGA) and DAPI were added post-fixation. The images correspond to

*Figure 5 continued on next page*

*Figure 5 continued*

a maximal projection of three or four slices of the z-stack encompassing the tunneling nanotubes (TNTs) indicated with yellow arrowheads. The scale bars correspond to 10 µm. (**B**) Graph of the percentage of TNT-connected cells in NT, CD46 AB, or CD9 AB treatment in WT cells. Mean percentages ± SD (N=3) are 30.4, 31.9 and 41.8, p-values resulting from statistical analysis are indicated above the brackets. (**C**) Top, schematic of the antibody treatment experiment after coculture of WT SH-SY5Y donor cells (DiD in yellow circles to stain the vesicles) and WT GFP-labeled acceptor cells, treated with control antibodies (CTR AB) or with antibodies anti-CD9 (CD9 AB) for an additional 2 hr. Graph represents the percentage of acceptor cells containing DiD vesicles of the cocultures of WT cells with CTR AB or CD9 AB treatment. Mean ± SD (N=3) are: CTR AB = 27.7 ± 2.48; CD9 AB = 36.8 ± 3.03. (**D**) Representative confocal images as in A except that CD81 KO cells were used. (**E**) Graph of the percentage of TNT-connected cells in NT, CD46 AB, or CD9 AB treatment in CD81 KO cells. Mean percentages ± SD (N=3) are 29.3, 28.5 and 41.4, p-values resulting from statistical analysis are indicated above the brackets. (**F**) Top, schematic of the antibody treatment experiment after coculture of CD81 KO SH-SY5Y donor cells (DiD in yellow circles to stain the vesicles) and WT GFP-labeled acceptor cells, treated with control antibodies (CTR AB) or with antibodies anti-CD9 (CD9 AB) for an additional 2 hr. Graph represents the percentage of acceptor cells containing DiD vesicles, mean percentages ± SD (N=3) are: CTR AB = 20.6 ± 1.67; CD9 AB = 20.6 ± 2.16.

The online version of this article includes the following figure supplement(s) for figure 5:

**Figure supplement 1.** CD9 vs. CTR antibody treatment in wild-type (WT) and CD81 knock-out (KO) cells.

---

actin chromobody-GFP content: 1. overlapping with phalloidin throughout the structure (fully green exemplified in the first row of *Figure 7B* pictures); 2. partially overlapping with phalloidin or 3. not present at all in the TNTs (exemplified in second and third rows, respectively). These different possibilities (summarized in the diagram of *Figure 7B*) could respectively correspond to TNTs whose iTNTs from donor cells reached the opposite cells (1), did not reach acceptor cells, or were non-open (2), and to TNTs growing only from acceptor cells (which were WT, (3)). As shown in *Figure 7C* graph, the percent of fully green TNT dropped from 69% in experiments with WT donor cells to 34% using CD81 KO donor cells, at the profit of partially green TNT, 'not green TNT' being minor in both cultures. These results suggested that in CD81KO cells, iTNT growth could be induced, but less iTNTs grow all the way to the acceptor cells, resulting in less transfer to acceptor cells. These results are consistent with CD81 favoring the full achievement of TNT formation, possibly playing a role in iTNT membrane docking to or fusion with opposing membrane whereas CD9 would stabilize TNT (see the working model in *Figure 7D*).

## Discussion
### The composition of TNTs is unique

Thanks to a newly established procedure in U2OS cells, we were able to reproducibly isolate a cellular fraction from cell bodies and from EVPs, and to analyze their content by mass spectrometry. The physical characteristics as well as the proteome analysis of this fraction, which is the TNT-microsomal-type/membrane proteome (called TNTome for simplicity) suggested that it could be enriched in TNTs with minor contaminations with other cell protrusions material. Therefore, its qualitative analysis could give valuable information on core components, traveling material or regulatory factors of TNTs, although we cannot rule out that some are specific to U2OS cells. Our results regarding the unique composition of TNTs were in accordance with those obtained by *Gousset et al., 2019*, who used a laser-captured microdissection approach combined with mass spectrometry to reveal the composition of various types of cellular protrusions in mouse CAD cells, including growth cones, filopodia, and TNTs. Interestingly, among the 190 proteins identified in *Gousset et al., 2019* in 2 samples from hCAD samples (enriched in TNTs), 101 were also found in TNTome (listed in *Supplementary file 7*), mostly corresponding to the most abundant cellular proteins. It is possible that compared to our approach, laser microdissection allowed a limited amount of material to be purified, and therefore few membrane proteins were identified.

TNTome is rich in membrane-associated proteins, as well as in proteins linked to the cytoskeleton, in particular to actin, as it was expected for these structures. TNTome is also abundant in ribonucleoproteins and translation-related factors (220 proteins, 19%). Some of them (as well as the proteins identified as nuclear in the databases) could be contaminants coming from cellular debris, however,

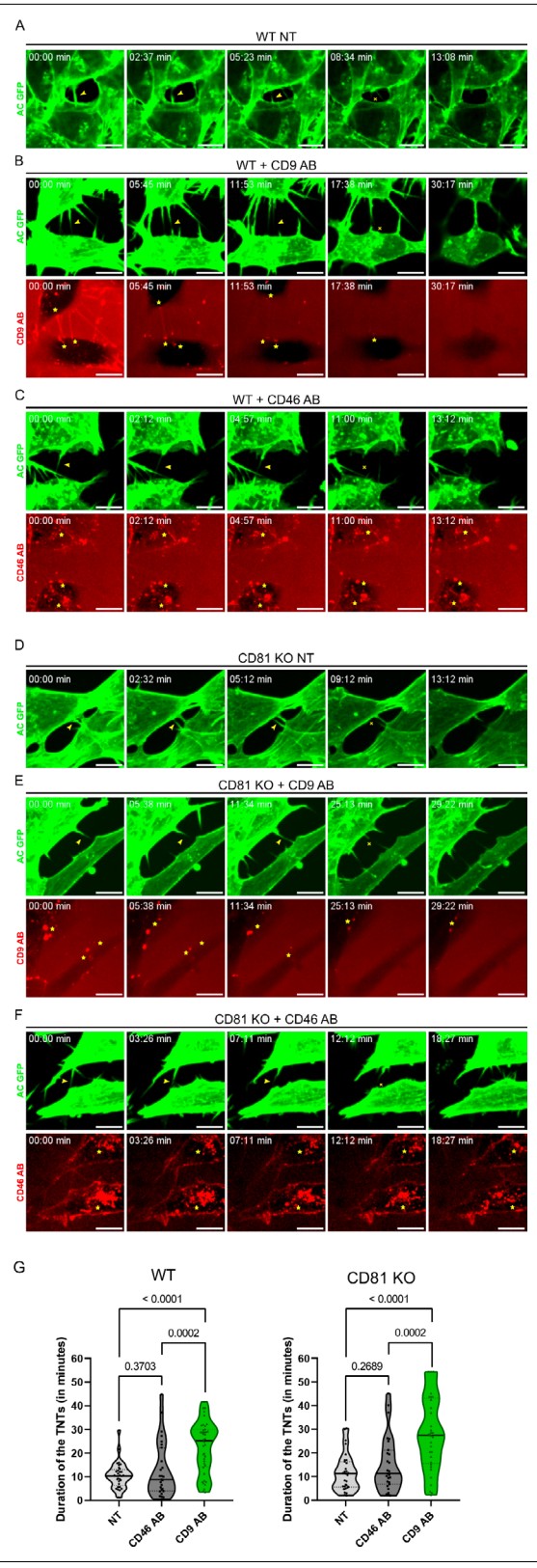

**Figure 6.** Stabilization of tunneling nanotubes (TNTs) by CD9 AB. (**A**) Representative snapshots of a TNT over time from *Figure 6—video 1*, corresponding to non-treated wild-type (WT) SH-SY5Y cells (WT NT). In A-F, green panels correspond to the signal of actin chromobody-GFP (AC GFP). Yellow arrowheads in the green panels are pointing to TNTs and the yellow crosses mark the breakage/dissociation of the TNT. Yellow stars in red panels mark

*Figure 6 continued on next page*

*Figure 6 continued*

the accumulation of CD9 AB (at the base/tip of the TNTs) or CD46 AB (intracellularly). Scale bars correspond to 10 μm. (**B**) Representative snapshots of TNTs over time from *Figure 6—video 2*, corresponding to WT SH-SY5Y cells treated with 10 μg/mL of CD9 specific antibody (WT + CD9 AB). Red panel corresponds to the signal of the CD9 antibody coupled to Alexa Fluor 568. (**C**) Representative snapshots of TNTs over time from *Figure 6— video 3*, corresponding to WT SH-SY5Y cells treated with 10 μg/mL of CD46 specific antibody (WT + CD46 AB), coupled to Alexa Fluor 594 (red panel). (**D**) Representative snapshots of a TNT over time from *Figure 6—video 4*, corresponding to non-treated CD81 KO SH-SY5Y cells (CD81 KO NT). (**E**) Representative snapshots of TNTs over time from *Figure 6—video 5*, corresponding to CD81 KO SH-SY5Y cells treated with 10 μg/mL of CD9 specific antibody (CD81 KO +CD9 AB), coupled to Alexa Fluor 568. (**F**) Representative snapshots of TNTs over time from *Figure 6—video 6*, corresponding to CD81 KO SH-SY5Y cells treated with 10 μg/mL of CD46 specific antibody (CD81 KO +CD46 AB) coupled to Alexa Fluor 594. (**G**) Average duration of TNTs in WT (left) or CD81 KO cells (right), measured by live imaging from three independent experiments, and represented in Violin plots (with the line at the median). Left: The mean lifetime of 33 TNTs in WT non-treated (NT) cells was 10.5 min (±5.59), and 12.8 min (±11.68) in 27 TNTs measured for WT cells treated with CD46 AB, while for WT cells treated with CD9 AB the average duration of the 40 TNTs measured was 22.6 min (±10.87). Right: mean lifetime of TNTs in CD81 KO non-treated (NT) cells, CD81 KO cells treated with CD46 AB, and CD81 KO cells treated with CD9 AB. The average lifetime of 31 TNTs in CD81 KO NT cells was 11.7 min (±7.78), 15.3 min (±11.70) in 27 TNTs in CD81 KO cells treated with CD46 AB, while the average duration of 33 TNTs in CD81 KO cells treated with CD9 AB was also 28.2 min (±15.66). Statistical analysis was performed using one-way Anova with Holm-Sidak's multicomparison test.

The online version of this article includes the following video(s) for figure 6:

**Figure 6—video 1.** Representative video of the stability measurement of tunneling nanotubes (TNTs) in wild-type (WT) SH-SY5Y cells non-treated with any antibody.

https://elifesciences.org/articles/99172/figures#fig6video1

**Figure 6—video 2.** Representative time-lapse microscopy video of tunneling nanotubes (TNTs) in SH-SY5Y WT cells as in *Figure 6—video 1*, except that cells were treated with 10 μg/mL of CD9-Alexa 568 antibody.

https://elifesciences.org/articles/99172/figures#fig6video2

**Figure 6—video 3.** Representative time-lapse microscopy video of tunneling nanotubes (TNTs) in SH-SY5Y wild-type (WT) cells treated with 10 μg/mL of CD46 antibody, as in *Figure 6—video 2*.

https://elifesciences.org/articles/99172/figures#fig6video3

**Figure 6—video 4.** Representative time-lapse microscopy video of tunneling nanotubes (TNTs) in SH-SY5Y CD81 KO cells non-treated with any antibody, as in *Figure 6—video 1*.

https://elifesciences.org/articles/99172/figures#fig6video4

**Figure 6—video 5.** Representative time-lapse microscopy video of tunneling nanotubes (TNTs) in SH-SY5Y CD81 KO cells treated with 10 μg/mL of CD9 antibody, as in *Figure 6—video 2*.

https://elifesciences.org/articles/99172/figures#fig6video5

**Figure 6—video 6.** Representative time-lapse microscopy video of tunneling nanotubes (TNTs) in SH-SY5Y CD81 KO cells treated with 10 μg/mL of CD46 antibody.

https://elifesciences.org/articles/99172/figures#fig6video6

this fraction should not be more than 10% of the total, based on nanoFCM results. Together with the fact that TNTs transfer mitochondria and lysosomes, it is possible that TNTs are used as a route to transfer RNAs alone or tethered to organelles by Annexins (several members in TNTome, including A11), G3BP1, or CAPRIN1 (both in the first quartile of TNTome *Liao et al., 2019*; *Lesnik et al., 2015*; *Cioni et al., 2019*) or that local translation happens along the TNT to fuel it with the required material. Transfer of mRNAs and microRNAs through TNTs has been described (*Haimovich et al., 2021*; *Kolba et al., 2019*) and could be one of the major functions of this kind of communication.

Regarding membranes, besides tetraspanins, cadherins, and integrins, some other plasma membrane TM proteins are of interest. Several amino acids and monocarboxylate transporters (respectively, SLC3A2, SLC1A5 in the first quartile, SLC1A4 and SLC7A1 in the fourth; and SLC16A1 in the second quartile) were found, suggesting together with the presence of mitochondria and of many metabolic enzymes that active metabolism takes place in TNTs. This may be necessary to generate local ATP to power TNT growth as previously suggested (*Garde et al., 2022*). Also, of great interest are SLC1A4 (neutral amino acid transporter) and STX7, which are the only TM proteins uniquely found in TNT and not EVP fraction. Overall, our results suggest that TNTs are different from EVPs, although

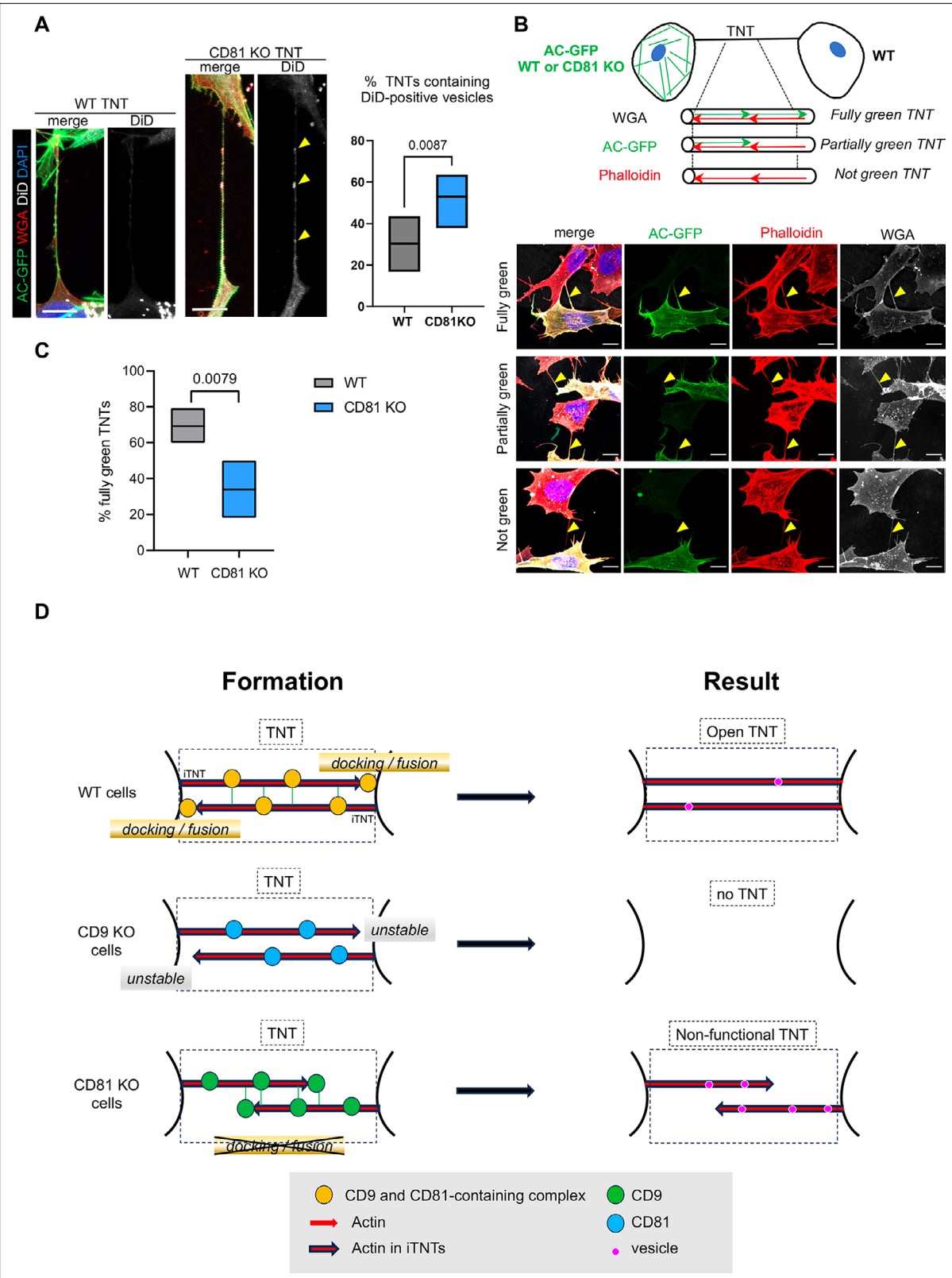

**Figure 7.** Completion of tunneling nanotubes (TNTs) by CD81. (**A**) DiD vesicles in TNTs from wild-type (WT) or CD81 KO cells. Actin chromobody-expressing SH-SY5Y cells, either WT or CD81 KO, were challenged with DiD for 30 min and fixed to preserve TNTs after 8 or 24 hr. After additional wheat germ agglutinin (WGA) staining, TNTs (above 10 µm in length) were identified by confocal microscopy using a WGA channel, and classified according to their content in DiD vesicles. Left is shown representative examples of long TNTs containing (projection of 12 slices of 0.19 µm) or not

*Figure 7 continued on next page*

*Figure 7 continued*

(projection of 7 slices) DiD-positive vesicles, from CD81 knock-out (KO) and WT cells respectively. Scale bars are 10 µm, and yellow arrowheads point to DiD-positive vesicles in the TNT. The floating bar graph on the right shows the percentage of TNTs that contain DiD-positive vesicles from six independent experiments (min to max results, line at means (30.3 and 53.0), medians are 33.4 and 53.7). Statistical analysis is the Mann-Whitney test, p-value is above the brackets. TNT number per experiment: 21, 23, 19 (8 hr of treatment), 6, 30, 41 (24 hr of treatment) for WT cells; 22, 17, 29 (8 hr of treatment), 17, 22, 29 (24 hr of treatment) for CD81KO cells. See also *Figure 7—videos 1–3*. (**B**) Coculture of actin chromobody-GFP-expressing cells (WT or CD81 KO) with WT cells. After fixation, cells were additionally labeled with WGA (Alexa 647, in white) and Phalloidin-Rhodamin (red). TNTs were identified by confocal imaging and classified as schematized above the diagrams: fully green when red and green signals overlapped throughout the structure (exemplified in the first row of pictures), partially or not green when green signal was interrupted or not present at all in the TNTs (exemplified in second and third rows respectively). First and third rows are maximal projections of six and seven slices (of 0.19 µm) encompassing the TNT of WT-WT coculture, the second row is a maximal projection of five slices of CD81 KO-WT coculture. Yellow arrowheads point to TNTs. Scale bars are 10 µm. (**C**) Quantification of the percentage of fully green TNTs in the cocultures of panel B, from five independent experiments (approximately 20 TNTs per condition and experiment, in total 91 and 92 analyzed TNTs from WT-WT and CD81 KO-WT cocultures respectively). Floating bars are min to max results, line at means (69.3 and 34%), medians are 66.7 and 27.8%, and statistical analysis was the Mann-Whitney test. (**D**) Working model of CD9/CD81 roles on TNT formation. TNTs (black dotted-line frame) are made up of iTNTs, supported by actin (red arrows). In WT cells, CD9 and CD81 are present on growing iTNT (probably together with additional interactors, indicated as a yellow circle), and CD9 stabilizes them. At the junction between iTNT and the opposing cell membrane, CD81 participates in membrane docking/ fusion, which results in the full completion of an open, functional TNT (right panel). When cells are treated with CD9AB, the active complexes are further enriched/stabilized, and more TNTs are stabilized. In CD9 KO cells, growing iTNTs are not stabilized, resulting in the decrease of TNT number. In CD81 KO cells, CD9-containing complexes (green circles) are still active for stabilization of iTNTs but anchoring of the iTNTs to opposite cell/fusion is no longer induced by CD81, resulting in apparent, but non-functional TNTs, in which vesicles (purple circles) are trapped.

The online version of this article includes the following video(s) for figure 7:

**Figure 7—video 1.** Time-lapse microscopy video of DiD-labeled vesicles (purple) passing from one cell to another through a tunneling nanotube (TNT) in SH-SY5Y WT cells expressing actin chromobody-GFP (green).

https://elifesciences.org/articles/99172/figures#fig7video1

**Figure 7—video 2.** Time-lapse microscopy video of DiD-labeled vesicles (purple, white arrowhead) entering from one cell into a tunneling nanotube (TNT), and next trapped into the TNT (yellow arrowhead) or at the junction of TNT with cell (white arrowhead) in SH-SY5Y CD81 KO cells expressing actin chromobody-GFP (green).

https://elifesciences.org/articles/99172/figures#fig7video2

**Figure 7—video 3.** Time-lapse microscopy video of DiD-labeled vesicles (purple) in SH-SY5Y CD81 knock-out (KO) cells expressing actin chromobody-GFP (green) culture.

https://elifesciences.org/articles/99172/figures#fig7video3

they share numerous factors. Further studying TNT proteins not present in EVPs will possibly allow us to identify TNT-specific markers.

To further validate our proteomic approach, we focused our attention on tetraspanins CD9 and CD81. Among the structures described to comprise at least one of these factors and that could potentially be contaminants of our TNT preparations, are midbodies. Midbodies have been shown to contain CD9 and its partners CD9-P1 and EWI-2 (*Addi et al., 2020*), and these structures could be copurified with TNTs, although in lower amounts possibly because cells were maintained in a serum-free medium for 24 hr before harvesting TNTs. Of note, only 13 TM proteins are common between TNTome and flemingsome (which has 29), including CD9, CD9P1, and EWI2, but also CADHF1 and 4, CD44, and Integrin α3. Some of these proteins could have a specific role in both TNTs and midbodies, like CD9 and its associated factors. Alternatively, some of these proteins, relatively abundant on plasma membranes, could be randomly present, independently of the specific structure that is analyzed. Importantly, no other specific markers of midbodies were detected in our MS data, including CRIK, CEP55, PLK1, PRC1, MKLP1, and 2, although they are expressed in U2OS cells (*Beck et al., 2011*; *Lundberg et al., 2010*) confirming that TNTs and midbodies have a different composition.

Among the cellular structures that depend on tetraspanin-enriched microdomains are also the recently described migrasomes, which are substrate-attached membrane elongated organelles formed on the branch points or the tips of retraction fibers of migrating cells which allow the release and transfer of cellular material in other cells (*Ma et al., 2015*). Although migrasomes have been described to be enriched in, and dependent on tetraspanins-enriched microdomains (*Huang et al., 2019*; *Huang et al., 2022*), they are fundamentally different from TNTs since they are attached to the substratum (and probably not collected during the procedure of TNT purification), and exhibit specific markers, identified by MS analysis, which are absent from TNTome (TSPAN 4 and 9, NDST1,

PIGK, CPQ, EOGT for example, see *Zhao et al., 2019*). Altogether, our TNT purification protocol and proteomic analysis have revealed that TNTs have a specific composition opening new avenues to understand how TNTs are formed and regulated.

## CD9 and CD81 regulate the formation of TNTs

TNTome in U2OS revealed that CD9 and CD81 are among the most abundant integral membrane proteins in TNTs (see *Supplementary file 3*, tab2). Consistent with previous results of overexpression (*Lachambre et al., 2014*), our data showed that they are also present in TNTs in other cell lines, especially SH-SY5Y cells, making them probable compulsory proteins of TNTs. Based on previously reported roles for these proteins, CD9 and CD81 were interesting candidates to play a role in TNT formation and/or function. Indeed, CD9 senses membrane curvature (*Dharan et al., 2022*), and several studies have reported a strong impact of these tetraspanins on membrane extensions (*Bari et al., 2011*), the most striking example being a profound modification of microvilli shape and distribution at the surface of CD9 KO oocytes (*Runge et al., 2007*). This may be due to their inverted cone-like structure which can induce membrane deformation in vitro (*Umeda et al., 2020*). In addition, both CD9 and CD81 have been shown to be involved in fusion, an important step in TNT formation following membrane extension; as positive regulators of sperm-egg fusion, but negative regulators of macrophage and muscle cell fusion (*Charrin et al., 2013*; *Rubinstein et al., 2006a*; *Rubinstein et al., 2006b*; *Kaji et al., 2002*; *Mascarau et al., 2023*). We show here that CD9 and CD81 are positive regulators of TNT function, since cell contact-dependent vesicle transfer in coculture experiments is decreased when CD9 or CD81 is lacking. Interestingly and despite their abundancy in EVPs, CD9 and/or CD81 absence did not affect transfer through secretion. These results confirmed recent data from the literature showing that CD9 and CD81 have minimal impact on the production or composition of EVs in MCF7 cells (*Fan et al., 2023*) and that the lack of CD9 does not impair the delivery of the content of EVs into recipient cells (*Tognoli et al., 2023*). Interestingly, EVs have been shown to regulate TNT formation in mesothelioma, breast cancer, or brain endothelial cell models (*Mahadik and Patwardhan, 2023*; *Thayanithy et al., 2014*; *Mentor and Fisher, 2022*). Although we cannot completely rule out a crosstalk between EVP and TNT formation or stability, our results together with the current data in the literature indicate that CD9 and CD81 deficiencies have a stronger impact on TNT function than on EVP production or function, suggesting that EVPs and TNTs are regulated independently of each other, as far as tetraspanins are concerned.

## Although CD9 and CD81 can partially compensate for each other, they act at different steps of TNT formation

Overall, our results indicate that CD9 and CD81 regulate cell-contact-mediated transfer, through different mechanisms. CD9 positively regulates the % of cells connected by TNTs, likely by stabilizing the TNTs. Indeed, short-term CD9 antibody treatment significantly increased TNT-connected cells and vesicle transfer, and this was associated with an increased TNT duration. CD9 antibody treatment also relocated CD9 and CD81 to TNT extremities, supporting the role of CD9-containing complexes in stabilizing TNTs maybe by favoring cis and/or trans interactions.

In contrast to CD9, CD81 regulates vesicle transfer without any effect on the number of TNT-connected cells, suggesting that these TNTs are less functional. Compared to WT cells, live cell imaging of CD81 KO cells showed that vesicles can still enter TNTs but do not efficiently transfer to the opposite cells, therefore, accumulating to some extent inside TNTs, as confirmed by the increased number of TNTs that contain vesicles after fixation. One possible explanation could be a reduced ability of CD81 KO TNTs to fuse with the opposing cell. However, we have shown that a GFP-labeled actin chromobody reached the opposing cell twice less frequently when expressed in CD81 KO cells than when expressed in WT cells, pointing again to an abnormal structure of these TNTs in the absence of CD81. The TNTs of SH-SY5Y cells are formed by a bundle of closely apposed individual tubes, iTNTs (*Sartori-Rupp et al., 2019*), originating from the two connected cells, which by confocal microscopy are detected as one single TNT, due to low resolution. The diminished percentage of TNTs fully labeled by the actin chromobody could, therefore, be explained by a decrease in the fraction of iTNTs originating from CD81 KO cells that reach or are stabilized at the opposing cell. Therefore, CD81 could act to facilitate anchoring of the growing iTNT membrane to the opposite membrane, ultimately enabling fusion and completion of fully open/functional TNTs.

Although CD81 does not seem to regulate the number of TNTs, several lines of evidence indicate that these two tetraspanins can compensate for one another in the formation/maintenance of TNTs and their function. First, the double KO of CD9 and CD81 reduced TNT-connected cells to levels lower than the CD9 single KO, indicating partially redundant roles for CD9 and CD81 in TNT biogenesis or stability. CD9 can also compensate for the lack of CD81 as its OE bypasses the requirement for CD81 for full vesicle transfer. This is reminiscent of the partial compensation by CD81 of the absence of CD9 on eggs during sperm-egg fusion (*Rubinstein et al., 2006a*; *Rubinstein et al., 2006b*; *Kaji et al., 2002*). In contrast, KO of CD9 on CD81 OE cells resulted in a significant decrease in the % of TNT-connected cells and vesicle transfer compared to CD81 OE alone. Furthermore, anti-CD9 antibody treatment of CD81 KO cells increased TNT number and duration to levels seen in WT cells, indicating that CD9 TNT-formation/stabilization capacity was independent of CD81. However, the same antibody incubation stimulates vesicle transfer in WT but not CD81 KO cells, suggesting that CD81 controls a different step. Based on these results, we propose the working model shown in *Figure 7D*, in which CD9 would act by stabilizing the interactions between iTNTs and bringing CD81 to the docking site between iTNT tip and opposite cell, allowing CD81 to participate in the anchoring of opposite membranes and finally in the opening of the channel. Recent work (*Huang et al., 2022*) proposed a role for TEM in forming a rigid ring impairing membrane damage to spread. Similarly, the specific CD9/CD81 TEM could somehow protect the membrane around the TNT site where fusion with the opposite cell occurs. More work in identifying specific TEM composition in TNTs would be needed to address this hypothesis.

In summary, in this study, we have shown that TNTs and EVPs are two cellular structures with partially overlapping composition, and that despite being part of both TNTs and EVPs, the tetraspanins CD9 and CD81 are fundamental regulators of TNT formation with complementary roles in the whole process of biogenesis of these structures. TNTome further analysis will be of great help to identify additional proteins, as part of the TEM or not, that could participate in TNT formation and regulation.

## Materials and methods
### Cell lines, lentiviral preparations, plasmids, and transfection procedures

U2OS cells were cultured at 37 °C in 5% CO2 in Dulbecco's Modified Eagle's Medium (DMEM + Glutamax, +4.5 g/l Glucose, +Pyruvate, Gibco), plus 10% fetal calf serum (FCS) and 1% penicillin/streptomycin (P/S). U2OS stably expressing H2B-GFP (Addgene 11680) and actin chromobody GFP (pAC-TagGFP from Chromotek) were obtained by transfection with Fugene HD according to manufacturer's instructions, followed by sorting of GFP-positive cells. GFP-CD9 expressing U2OS cells were obtained by lentiviral transduction as below, followed by limiting dilution to obtain a clone. SH-SY5Y human neuroblastoma cells (gift from Simona Paladino, Department of Molecular Medicine and Medical Biotechnology, University of Naples Federico II, Naples, Italy) were cultured at 37 °C in 5% CO2 in RPMI-1640 (Euroclone), plus 10% FCS and 1% P/S.

For the lentiviral preparations, human HEK 293T cells were cultured at 37 °C in 5% CO2 in DMEM (ThermoFisher), with 10% FCS and 1% P/S. Cells were plated one day before transfection at a confluency of around 70%. Transfection of the different plasmids was made in a ratio 4:1:4 using lentiviral components pCMVR8, 74 (Gag-Pol-Hiv1), and pMDG2 (VSV-G) vectors and the plasmid of interest, respectively using FuGENE HD (Promega) according to manufacturer's protocol. CRISPR lentiviral plasmids containing the sequences for CD9: GAATCGGAGCCATAGTCCAA and CD81: AGGAATCCCAGTGCCTGCTG were selected using the CRISPR design tool available at the Broad Institute (https://portals.broadinstitute.org/gpp/public/analysis-tools/sgrna-design). The corresponding guide DNA sequences were cloned into the lentiCRISPRv2 plasmid (#52961; Addgene) according to the instructions of the Zhang laboratory (https://www.addgene.org/52961/). The viral particles were collected and concentrated using LentiX-Concentrator (TakaraBio) after 48 hr. To KO CD9 and/or CD81, SH-SY5Y were plated the day before the infection at a confluency of around 70% and the next day the lentivirus was added to the cells. 24 hr later the medium with the lentiviruses was removed and cells were selected with RPMI + 1 µg/mL of puromycin for 2 days. After this, cells were splitted 1:5 and reinfected for 24 hr. After these 24 hr, the medium was replaced with RPMI + 1 µg/mL of puromycin for 10 days changing the medium every 2–3 days. Finally, cells were tested for the absence of expression

of CD9 and/or CD81 by western blot (WB). These cells were used to generate Actin chromo body-GFP expressing cells, by transfection of the plasmid (pAC-TagGFP from Chromotek) using Lipofectamine 2000 according to manufacturer's instructions, next sorted to enrich in GFP-positive cells, which were finally cloned by limiting dilution.

To obtain clones that overexpress CD9 or CD81, SH-SY5Y were plated the day before at a confluency of around 70% and next day cells were transfected with the corresponding plasmid using Lipofectamine 2000 (Invitrogen) following the manufacture recommendations and 72 hr post-transfection cells selected with 350 µg/mL of Hygromicin B (Gibco) for 5 days, changing the medium every 2 days and then with 50 µg/mL of Hygromicin B for 10 days more. The CD9 or CD81 OE clones were obtained by limiting dilution and tested for expression of CD9 or CD81 by immunofluorescence and WB. All cell lines were regularly tested for the absence of mycoplasma contamination.

## TNT and EVP preparation

Two million U2OS cells were plated in 75 cm$^2$ flasks for 24 hr (eight flasks per point), next complete medium was replaced by a medium without FCS for an additional 24 hr. For EVP preparation, conditioned medium was collected, centrifuged twice at 2000 g to remove cells, concentrated 10-fold on Vivaspin 20 (MWCO 10kD, Cytiva), and next submitted to ultracentrifugation in a Beckman MLS50 rotor at 10,000 g for 30 min at 4 °C. Supernatant was collected and centrifuged at 100,000 g for 70 min at 4 °C, resulting pellet was resuspended in PBS and centrifuged again at 100,000 g for 70 min at 4 °C. Pellets were used for electronic microscopy, Mass Spectrometry, or solubilized in 2 x Laemmli for WB analysis.

For TNT preparation, cell cultures after removing the conditioned medium were washed carefully with PBS, next 2 ml of PBS was added to each flask, which was left on an oscillating shaker for 5 min before being shaken (30 s horizontally, and four times 30 s by banging them vigorously). PBS from all flasks was drained, collected, and centrifuged twice at 2000 g, and filtered on a 0.45 µM syringe filter (Corning) to remove detached cells, next submitted to ultracentrifugation at 100,000 g for 70 min at 4 °C. Pellets were used for electronic microscopy, Mass Spectrometry, or WB. After collecting EVPs and TNTs from cell cultures, cells were harvested in PBS, and cell extracts were prepared in Tris 50 mM pH 7.4, NaCl 300 mM, MgCl2 5 mM, Triton 1% with protease inhibitors (complete mini, Roche).

## Negative staining and transmission electron microscopy

EVP or TNT pellets were resuspended in 50 ml of PBS. Four microliters of each sample were spotted on a carbon-coated grid-primarily glow discharged and incubated at room temperature for 1 min. Uranyl acetate (2%) in water was used to contrast the grids and incubated for 1 min. The grid was then dried and observed under 120 kV using a Tecnai microscope (Thermo Fisher Scientific) and imaged using a 4000 by 4000 Eagle camera (Thermo Fisher Scientific). To quantify EVP diameter, vesicles were segmented manually using the Quick selection tool of PHOTOSHOP v23.5.5 (Adobe Systems, San Jose, CA). Spot detector under ICY software (https://icy.bioimageanalysis.org/) saved the surface area of the segmented vesicles to a .excel file. The vesicle size d was calculated from the surface area A using $d = \sqrt{4A/\pi}$, thereby assuming that vesicles are spherical. For TNTs, ROIs were defined under ICY, and first and second diameters were used for diameter and length, respectively.

## Nano-flow cytometry

The size and number of exosomes and TNT particles were identified by Nano-flow cytometry (NanoFCM). NanoFCM is applicable when the refractive index of input samples is the same or similar to that of silica particles. The standard working curve of scattering light intensity was established using a silica standard sphere. EVPs and TNT particles were isolated from 1 flask of culture following the protocol above except for the ultracentrifugation steps. The particle size distribution of samples was measured based on the scattering intensity.

## Mass spectrometry
### Digestion of TNT and EVPs samples

Protein pellets were dissolved in urea 8 M, Tris 50 mM pH 8.0, TCEP (tris(2-carboxyethyl)phosphine) 5 mM, and SDS (Sodium Dodecyl Sulfate) 2%. SDS was removed using a methanol/ chloroform/ water extraction. Briefly, 3 V of ice-cold Methanol was added to the sample and then mixed. 2 V of ice-cold

chloroform was added and mixed. Then 3V of ice cold water was added and mixed. Samples were spinned for 3 min at 5000 g at 4 °C. Proteins at the organic/inorganic interface were kept and washed three times in ice-cold methanol. Protein pellets were dissolved in Guanidine 1 M, TCEP 5 mM, Chloroacetamide 20 mM, and Tris 50mM pH8.0 and samples were heated 5 min at 90 °C before digestion in a mix of 500 ng of LysC (Promega) at 37 °C for 2 hr. Dilution with Tris 50 mM was done before the addition of trypsin 500 ng (Promega) and digestion at 37 °C for 8 hr. Digestion was stopped by adding 1% final of formic acid. Peptides were purified using a C18-based clean-up standard protocol done using Bravo AssayMap device.

## LC-MS/MS analysis of TNT and EVPs

LC-MS/SM analysis of digested peptides was performed on an Orbitrap Q Exactive Plus mass spectrometer (Thermo Fisher Scientific, Bremen) coupled to an EASY-nLC 1200 (Thermo Fisher Scientific). A home-made column was used for peptide separation ($C_{18}$ 50 cm capillary column picotip silica emitter tip (75 µm diameter filled with 1.9 µm Reprosil-Pur Basic $C_{18}$-HD resin, (Dr. Maisch GmbH, Ammerbuch-Entringen, Germany))). It was equilibrated and peptides were loaded in solvent A (0.1% FA) at 900 bars. Peptides were separated at 250 nl.min$^{-1}$. Peptides were eluted using a gradient of solvent B (ACN, 0.1% FA) from 3% to 22% in 140 min, 22 to 42% in 61 min, 42 to 60% in 15 min (total length of the chromatographic run was 240 min including high ACN level step and column regeneration). Mass spectra were acquired in data-dependent acquisition mode with the XCalibur 2.2 software (Thermo Fisher Scientific, Bremen) with automatic switching between MS and MS/MS scans using a top 10 method. MS spectra were acquired at a resolution of 70000 (at $m/z$ 400) with a target value of $3 \times 10^6$ ions. The scan range was limited from 400 to 1700 $m/z$. Peptide fragmentation was performed using higher-energy collision dissociation (HCD) with the energy set at 26 NCE. Intensity threshold for ions selection was set at $1 \times 10^6$ ions with charge exclusion of z=1 and z>7. The MS/MS spectra were acquired at a resolution of 17500 (at $m/z$ 400). Isolation window was set at 2.0 Th. Dynamic exclusion was employed within 35 s.

All Data were searched using MaxQuant (version 1.6.6.0) using the Andromeda search engine (*Tyanova et al., 2016*) against a human reference proteome (75088 entries, downloaded from Uniprot on the October 29, 2020).

The following search parameters were applied: carbamidomethylation of cysteines was set as a fixed modification, and oxidation of methionine and protein N-terminal acetylation were set as variable modifications. The mass tolerances in MS and MS/MS were set to 5 ppm and 20 ppm, respectively. Maximum peptide charge was set to 7 and 5 amino acids were required as minimum peptide length. At least two peptides (including 1 unique peptide) were asked to report a protein identification. A false discovery rate of 1% was set up for both protein and peptide levels. iBAQ value was calculated. The match between runs features was allowed for biological replicate only.

## Data analysis

Quantitative analysis was based on pairwise comparison of protein intensities. Values were log-transformed (log2). Reverse hits and potential contaminants were removed from the analysis. Proteins with at least two peptides were kept for further statistics. Intensity values were normalized by median centering within conditions (normalizeD function of the R package DAPAR *Wieczorek et al., 2017*). Remaining proteins without any iBAQ value in one of both conditions have been considered as proteins quantitatively present in a condition and absent in the other. They have, therefore, been set aside and considered as differentially abundant proteins. Next, missing values were imputed using the impute.MLE function of the R package imp4p (https://rdrr.io/cran/imp4p/man/imp4p-package.html). Statistical testing was conducted using a limma t-test thanks to the R package limma (*Pounds and Cheng, 2006*). An adaptive Benjamini-Hochberg procedure was applied to the resulting p-values thanks to the function adjust.p of R package cp4p (*Smyth, 2004*) using the robust method described in *Giai Gianetto et al., 2016* to estimate the proportion of true null hypotheses among the set of statistical tests. The proteins associated to an adjusted p-value inferior to an FDR level of 1% have been considered as significantly differentially abundant proteins.

## Bioinformatic analysis and data mining

Twelve replicates were done for the discovery of TNT proteins. Nine over 12 were kept as being part of the TNT. The 1177 resulting proteins were sorted by quartiles according to their iBAQ value. For each of the four lists, protein composition was depicted using the ProteoMap tool (*Liebermeister et al., 2014*) to visualize their weighted GO organization and protein contribution for each GO term. Also, a DAVID analysis (*Sherman et al., 2022*) was done for each quartile of proteins. Functional charts and clusters were used to describe the dataset. Protein networks were visualized using STRING (*Szklarczyk et al., 2021*) and Cytoscape (https://cytoscape.org/).

## Resource availability

The mass spectrometry proteomics data have been deposited to the ProteomeXchange Consortium via the PRIDE (*Rustom et al., 2004*) partner repository with the dataset identifier PXD033089.

## Sample preparation for TNT imaging

SH-SY5Y or U2OS-derived cells were trypsinized and counted, and 100,000 or 30,000 cells, respectively were plated on coverslips. After O/N culture, cells were fixed with specific fixatives to preserve TNT (*Abounit et al., 2015*), first with Fixative 1 (2% PFA, 0.05% glutaraldehyde, and 0.2 M HEPES in PBS) for 15 min at 37 °C, then with fixative 2 (4% PFA and 0.2 M HEPES in PBS) for 15 min at 37 °C. After fixation, cells were washed with PBS, and membranes were stained with conjugated wheat germ agglutinin (WGA)-Alexa Fluor (1:300 in PBS, Invitrogen) or Phalloidin-Texas-red (1:300 in PBS, Invitrogen) and DAPI (1:1000) (Invitrogen) at room temperature 15 min. After gently washing three times with PBS, samples were mounted on glass slides with Aqua PolyMount (Polysciences, Inc). Every different SH-SY5Y cell type (WT, tetraspanin KO, or tetraspanin OE) was prepared in the exact same conditions.

## Quantification of the percentage of TNT-connected cells (also referred to as TNT counting or TNT number)

For U2OS-derived cells, a mild trypsinization was used to be able to visualize connections (including TNTs) between cells in the same flasks used for TNT preparation. Cells were incubated for 4 min at 37 °C with a 100-fold dilution of 0.05% Trypsin-EDTA solution (Gibco), next cells were fixed as above, and phase images were acquired using Incucyte system (Essen Bioscience, 20 x magnification). For SH-SY5Y cells grown on coverslips, multiple random Z-stack images of different points on the samples were acquired using an inverted laser scanning confocal microscope LSM 700 or 900 (Zeiss) controlled by the Zen software (Zeiss). The images were analyzed according to the morphological criteria of TNTs: structures that connect two distant cells and that are not attached to the substratum. First slices were excluded from the analysis, and only connections in the middle and upper stacks were considered. Cells that have TNTs between them were marked as cells connected by TNTs, and the number of these cells was compared to the total number of cells in the sample, giving the percentage of cells connected by TNTs. For both cell types, the analysis was performed using ICY software (https://icy.bioimageanalysis.org/), by using the 'Manual TNT annotation' plugin. In each experiment, at least 200 or more cells were analyzed in every condition. The images were adjusted and processed with the ImageJ (https://imagej.net/) or Icy softwares.

## Co-culture assay (DiD transfer assay) and flow cytometry analysis

DiD transfer assays have been described elsewhere (*Jacquemet et al., 2019*). Briefly, the co-culture consisted of two distinctly labeled cell populations: a first population of cells (donors) was treated with Vybrant DiD (dialkylcarbocyanine), a lipophilic dye that stains vesicles, at 1:1000 (Thermo Fisher Scientific) in complete medium for 30 min at 37 °C (Life Technologies), the cells were then trypsinized and mixed in a 1:1 ratio with a different cell population (acceptors) of different color to distinguish them from donors (usually expressing GFP) and the co-culture was incubated O/N.

In the case of the co-cultures with KO, OE, or OE +KO of CD9, and CD81, 400,000 donor cells were mixed with 400,000 acceptor cells on six-well plates for analysis by flow cytometry. After O/N culture, cells were trypsinized, passed through a cell strainer to dissociate cellular aggregates, and fixed with 2% PFA in PBS. Finally, these cells were passed through the CytoFLEX S Flow Cytometer (Beckman Coulter) under the control of the CytoExpert Acquisition software. The data were analyzed

with FlowJo software following a similar strategy for all experiments: first, the samples were gated to exclude cellular debris by plotting the area obtained with the side scatter (SSC-A) and the area obtained with the forward scatter (FSC-A) obtaining all the cells in the sample. Second, within this previous gate, the sample was gated to exclude cell doublets, plotting the width obtained with side scatter (SSC-W) and the area obtained with forward scatter (FSC-A) thus obtaining the singlets. Finally, within the singlet gate, the co-culture was gated using GFP and DiD expression, resulting in four quadrants delimiting double-negative, GFP-positive, DiD-positive, and double-positive populations. The % of acceptor cells receiving DiD vesicles was obtained by calculating the percentage of acceptor cells with labeled vesicles out of the total number of acceptor cells.

In the case of the co-culture of the CD9 AB treatment, 50.000 donor cells were co-cultured with 50.000 acceptor cells on coverslips. Results were analyzed by microscopy as described above, and results were obtained by semi-quantitative analysis using ICY software (http://icy.bioimageanalysis.org/) by calculating the percentage of acceptor cells with labeled vesicles out of the total number of acceptor cells. In each experiment, at least 100 recipient cells per condition were counted. Image montages were built afterward in ImageJ software.

In all co-cultures, a control of the transfer by secretion was performed. DiD-loaded donor cells were seeded alone (800,000 cells for flow cytometry and 100,000 cells for microscopy) and cultured for 24 hr. Next, the supernatant from these cells was centrifuged and added to the acceptor cells that had been seeded on the previous day under the same conditions as the donors, these acceptor cells were cultured for an additional 24 hr and next fixed and analyzed in the same way as above mentioned.

## CD9 and CD46 antibody (AB) treatments

Both TNT counting and vesicle transfer in the CD9 or CD46 AB treatment were done in the same way as described, with an additional step after 24 hr of coculture consisting of the incubation of the cells with either Goat anti-rabbit Alexa Fluor 305 (Thermo Fisher ref: A21068-) control antibodies (CTR AB) or anti-CD9 (TS9, coupled to Alexa 568) or anti-CD46 (coupled to Alexa 594) antibodies at a concentration of 10 µg/mL in RPMI medium for 2 or 3 hr. Subsequently, the cells were fixed and submitted to immunofluorescence.

## Live imaging microscopy

Time-lapse microscopy imaging was performed on an inverted spinning disk microscope (Eclipse Ti2 microscope system, Nikon Instruments, Melville, New York, USA) using 60X1.4 NA CSU oil immersion objective lens and laser illumination 488 and 561, with optical sections of 0.4 µm. During image acquisition, 37 °C temperature, 5% $CO_2$, $O_2$, and humidity were controlled with a small environmental chamber (Okolab). Cells were plated in Ibidi µ-slide with a glass bottom. Image processing and movies were realized using ImageJ/Fiji software.

## Immunofluorescence

For immunofluorescence, 100.000 SH-SY5Y cells or 30,000 U2OS cells were seeded on glass coverslips and after O/N culture they were fixed with 4% paraformaldehyde (PFA) for 15 min, quenched with 50 mM $NH_4Cl$ for 10 min and blocked in 2% BSA in PBS for 20 min. Primary antibodies mouse anti-CD9 IgG1 (TS9), anti-CD81 IgG2a (TS81), anti-ITGB1 (β1-vjf, IgG1) or anti-CD151 (TS151, IgG1) were previously described (*Arduise et al., 2008*) and were used in 2% BSA in PBS during 1 hr. Other primary antibodies used after permeabilization were in 0.05% saponin, 0.2%-containing PBS: mouse anti-Vinculin (Sigma V9264, 1:500), rabbit anti-paxillin (Santa-Cruz, sc-5574, 1:1000), mouse anti-GM130 (BD 610823, 1:1000). After 3 washes of 10 min each with PBS, cells were incubated with each corresponding Alexa Fluor-conjugated secondary antibody (Invitrogen) at 1:1000 in 2% BSA in PBS for 1 hr. Specifically, the secondary antibodies used were goat anti-mouse with epitope IgG1 Alexa Fluor 488 for CD9 (Invitrogen ref: A21121) and goat anti-mouse with epitope IgG2a Alexa Fluor 633 for CD81 (Invitrogen ref: A21136). For the experiments showing the actin cytoskeleton, cells were labeled with Phalloidin- Rhodamine (Invitrogen) in the same mix and conditions as the secondary antibodies. Then, cells were washed three times for 10 min each with PBS, stained with DAPI, and mounted on glass slides with Aqua PolyMount (Polysciences, Inc). Images were acquired with a confocal microscope LSM700 or 900 (Zeiss) and processed with the ImageJ or Icy software.

## Western blot

For Western blot SH-SY5Y cells were lysed with lysis buffer composed of 150 mM NaCl, 5 mM EDTA, 20 mM Tris, pH 8.0, 1% Triton X100 with Roche *cOmplete Protease Inhibitor Cocktail*. Protein concentration was measured by a Bradford protein assay (Bio-Rad). 20 µg of protein were submitted to PAGE (in non-reducing conditions for tetraspanins and ADAM10) followed by western blot analysis. Primary antibodies used for Western blot were: mouse anti-CD9 IgG1 (TS9, 1:1000), mouse anti-CD81 IgG2a (TS81, 1:1000), mouse anti-CD63 (1:1000), mouse anti ADAM10 (11G2, 1:1000) (*Charrin et al., 2001*; *Arduise et al., 2008*), rabbit anti-α-GAPDH (Sigma ref: G9545, 1:1000), mouse anti-actin (MP Biomedicals, clone C4, 1:1000), mouse anti-α-tubulin (Sigma ref: T9026, 1:2000), mouse anti-GM130 (BD transduction Laboratories, 610823, 1:1000), rabbit ITGB1, ITGB4, ITGA4 (from integrin Ab Sampler kit, Cell signaling ref: 4749, 1:1000), rabbit anti-EGFR (Cell Signaling ref: 4267, 1:1000), rabbit anti-CX43 (Sigma ref: C6219, 1:3000), mouse anti-ANXA2 (Proteintech ref: 66035, 1:2000), mouse anti-Alix (Biorad MCA2493, 1:1000), rabbit anti-Calnexin (Enzo SPA-860, 1:1000).

## Statistical analysis

Statistical analysis of experiments concerning the TNT counting and the DiD transfer assay are described elsewhere (*Pinto et al., 2021*). Briefly, the statistical tests were applied using either a logistic regression model computed using the 'glm' function of R software (https://www.R-project.org/) or a mixed effect logistic regression model using the lmer and lmerTest R packages, applying a pairwise comparison test between groups.

All graphs shown in this study have been made with GraphPad Prism version 9.

The numerical data used in all figures are included in *Supplementary file 8*.

## Acknowledgements

We acknowledge the Center for Translational Science (CRT)-Cytometry and Biomarkers Unit of Technology and Service (CB UTechS), in particular PH Commère, as well as G Pehau-Arnaudet and A Mallet from the Ultrastructural Bioimaging (UBI) facility. Thanks to Dr. S Charrin for helpful discussion, Dr. S Lebreton, and R Chakraborty for critical reading of the manuscript, and all the members of the UTRAF unit for their support. We are endebted to the late Ms Marguerite MICHEL whose bequest to Institut Pasteur has made this project possible. We are grateful for financial support to CZ from Institut National du Cancer (PLBIO18-103), Association France Alzheimer (AAP PFA 2021 #6156), Equipe Fondation Recherche Médicale (FRM EQU202103012692), and Agence Nationale de la Recherche (ANR-20-CE13-0032 to CZ and ANR-21-CE35-0007 to MM). We are grateful for support for equipment from the French Government Programme Investissements d'Avenir France BioImaging (FBI, N° ANR-10-INSB-04–01) and the French gouvernement (Agence Nationale de la Recherche) Investissement d'Avenir programme, Laboratoire d'Excellence 'Integrative Biology of Emerging Infectious Diseases' (ANR-10-LABX-62-IBEID).

## Additional information

### Funding

| Funder | Grant reference number | Author |
|---|---|---|
| Institut National Du Cancer | PLBIO18-103 | Chiara Zurzolo |
| Fondation pour la Recherche Médicale | FRM EQU202103012692 | Chiara Zurzolo |
| Agence Nationale de la Recherche | ANR-20-CE13-0032 | Chiara Zurzolo |
| Agence Nationale de la Recherche | ANR-21-CE35-0007 | Mariette Matondo |
| Association France Alzheimer | AAP PFA 2021 #6156 | Chiara Zurzolo |

| Funder | Grant reference number | Author |
|---|---|---|

The funders had no role in study design, data collection and interpretation, or the decision to submit the work for publication.

## Author contributions

Roberto Notario Manzano, Conceptualization, Formal analysis, Methodology, Writing – original draft, Writing – review and editing; Thibault Chaze, Formal analysis, Methodology, Writing – review and editing; Eric Rubinstein, Methodology, Writing – review and editing; Esthel Penard, Formal analysis, Methodology; Mariette Matondo, Funding acquisition, Methodology, Writing – review and editing; Chiara Zurzolo, Conceptualization, Supervision, Funding acquisition, Writing – review and editing; Christel Brou, Conceptualization, Formal analysis, Supervision, Methodology, Writing – original draft, Writing – review and editing

## Author ORCIDs

Roberto Notario Manzano ⓘ https://orcid.org/0000-0002-2435-2938
Thibault Chaze ⓘ https://orcid.org/0000-0002-3615-7021
Eric Rubinstein ⓘ https://orcid.org/0000-0001-7623-9665
Esthel Penard ⓘ https://orcid.org/0000-0002-4442-7503
Mariette Matondo ⓘ https://orcid.org/0000-0003-3958-7710
Chiara Zurzolo ⓘ http://orcid.org/0000-0001-6048-6602
Christel Brou ⓘ https://orcid.org/0000-0003-0229-8202

Reviewer #1 (Public Review): https://doi.org/10.7554/eLife.99172.2.sa1
Reviewer #2 (Public Review): https://doi.org/10.7554/eLife.99172.2.sa2
Reviewer #3 (Public Review): https://doi.org/10.7554/eLife.99172.2.sa3
Author response https://doi.org/10.7554/eLife.99172.2.sa4

---

# Additional files

## Supplementary files

• Supplementary file 1. The full TNTome (1177 proteins), from 12 independent samples, proteins were conserved when present in more than 9 replicates, ranked in four quartiles (from higher to lower mean iBAQ) represented in different colors (orange Q1, green Q2, pink Q3, blue Q4). iBAQ of each sample is indicated for each protein. GO terms are indicated in the last columns.

• Supplementary file 2. Cytoskeleton-associated proteins of the TNTome. Proteins identified by GO term analysis (cellular components), except those associated to proteasome, RNA and mitochondria, are ranked according to their quartile assignment (orange Q1, green Q2, pink Q3, blue Q4).

• Supplementary file 3. Membrane-related proteins identified by GO term analysis (cellular components). Mitochondrial and other organelles membrane proteins have been discarded from GO analysis. Tab 1 lists all membrane and membrane-associated proteins, tab 2 only the integral membrane proteins, ranked according to their quartile assignment (orange Q1, green Q2, pink Q3, blue Q4) and from more abundant to less abundant. Integrins, Ephrin receptors, Cadherins, and tetraspanin-related proteins are highlighted, respectively in orange, red, dark green and yellow as indicated in tab2.

• Supplementary file 4. Comparison of TNTome and Integrin adhesion complexes shows the common elements in integrin adhesion complexes (2240 proteins according to *Horton et al., 2015*) and in TNTome: 765 proteins listed in alphabetical order of the gene name (yellow background). On the right (blue background) are the 413 elements included exclusively in TNTome. Tab2 shows the 26 common elements in consensus adhesome (*Horton et al., 2015*) and TNTome.

• Supplementary file 5. TNT-only proteins. Tab 1 (total) shows the 174 proteins present in TNT preparations and absent from extracellular vesicles and particles (EVPs). Tab2 (constitutive) shows the 89 tunneling nanotube (TNT)-only proteins without proteins described as mitochondrial, nuclear, ER or RNA-related. In yellow background are cytoskeleton-related proteins (20%).

• Supplementary file 6. Overlapping proteins between tunneling nanotubes (TNTs) and extracellular vesicles and particles (EVPs). Tab1 (TOT TNT >EVP) shows the proteins more abundant in TNTs compared to EVPs, cleaned of nuclear, mitochondrial or RNA-related described proteins in tab2

(TNT >EVP). In yellow background are cytoskeleton-related proteins (29%). Tab7 shows the proteins present in EVPs and not in TNTs (except for the 10 proteins in grey background that were in the full TNTome).

• Supplementary file 7. Common proteins between TNTome and protrusions from hCAD cells described in *Gousset et al., 2019*. The 190 proteins present in two samples of hCAD (mouse CAD cells treated with H2O2) were converted to their human ortholog, next compared to the 1177 proteins of *Supplementary file 1*.

• Supplementary file 8. Excel spreadsheet containing, in separate sheets, the underlying numerical data and statistical analysis for Figure panels 1C, 1E, 3C, 3E, 3F, 3G, 4B, 4C, 5B, 5C, 5E, 5F, 6G, 7A, 7C, S1D, S1E, S1G, S1H, S3C, S5B, S5D, S5F, S6B, S6C, S6E, S6F, S7A, S7B.

• MDAR checklist

### Data availability

The mass spectrometry proteomics data have been deposited to the ProteomeXchange Consortium via the PRIDE partner repository with the dataset identifier PXD033089. *Supplementary file 8* contains numerical data and statistical analysis for all figures. Source data files have been provided for all WBs.

The following dataset was generated:

| Author(s) | Year | Dataset title | Dataset URL | Database and Identifier |
|---|---|---|---|---|
| Brou C | 2024 | Proteomic landscape of tunneling nanotubes reveals CD9 and CD81 tetraspanins as key regulators | https://www.ebi.ac.uk/pride/archive/projects/PXD033089 | PRIDE, PXD033089 |

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

# Appendix 1

## Appendix 1—key resources table

| Reagent type (species) or resource | Designation | Source or reference | Identifiers | Additional information |
|---|---|---|---|---|
| Cell line (*Homo sapiens*) | U2OS | ATCC | HTB-96 | From osteosarcoma |
| Cell line (*H. sapiens*) | SH-SY5Y | gift from Simona Paladino (Department of Molecular Medicine and Medical Biotechnology, University of Naples Federico II, Naples, Italy) | | From neuroblastoma |
| Cell line (*H. sapiens*) | HEK 293T | | | Used for lentiviral production |
| Transfected construct (*H. sapiens*) | H2B-GFP | Addgene | #11680 | Human histone H2B (*H2BC11* Human) |
| Transfected construct (*Alpaca*) | AC-GFP | Chromotek | pAC-TagGFP | mammalian expression vector encoding the cytoskeleton marker Actin-VHH fused to green fluorescent protein TagGFP2 |
| Transfected construct (*H. sapiens*) | GFP-CD9 | This paper: The TRIPΔ3-EF1α-CD9 plasmid was constructed by inserting the human CD9cDNA sequence in the TRIPΔ3-EF1α vector (A generous gift from Anne Dubart-Kupperschmitt); *Sirven et al., 2001* | | Lentiviral vector backbone |
| Transfected construct (*HIV-1*) | pCMVR8,74 | Addgene | #22036 | From Trono lab, 2nd generation lentiviral packaging plasmid. |
| Transfected construct (*VSV*) | pMDG2 | Addgene | #12259 | From Trono lab, VSV-G envelope expressing plasmid |
| Transfected construct (*H. sapiens*) | lentiCRISPRv2 targeting human CD9 | CD9 target: GAATCGGAGCCATAGTCCAA | lentiCRISPRv2: Addgene #52961 | Lentiviral vector |
| Transfected construct (*H. sapiens*) | lentiCRISPRv2 targeting human CD81 | CD81 target: AGGAATCCCAGTGCCTGCTG | lentiCRISPRv2: Addgene #52961 | Lentiviral vector |
| Antibody | Mouse monoclonal anti-CD9 IgG1 TS9 | *Le Naour et al., 2006* | TS9 Diaclone: #857.750.000 | IF (1:1000), WB (1:1000) |
| Antibody | Mouse monoclonal anti-CD46 | *Lozahic et al., 2000* | | Live (1/100) |
| Antibody | Secondary goat polyclonal antibodies-Alexa fluor | In vitrogen (Thermo Fisher Scientific) | Various references | IF (1/1000) |
| Antibody | Mouse monoclonal anti-CD81 IgG2a | *Charrin et al., 2001* | Diaclone: # 857.780.000 | IF (1:1000), WB (1:1000) |
| Antibody | Mouse monoclonal anti ITGB1 IgG1 | *Le Naour et al., 2006* | | IF (1:500) |
| Antibody | Mouse monoclonal anti-CD151 IgG1 | *Charrin et al., 2001* | Merck Millipore # MABT59 | IF (1:500) |
| Antibody | Mouse monoclonal anti-vinculin | Sigma | #V9264 | IF (1/1000) |
| Antibody | Rabbit polyclonal anti-paxillin | Santa-Cruz | #Sc-5574 | IF (1/1000) |
| Antibody | Purified Mouse monoclonal anti-GM130 IgG1 | BD transduction laboratories | BD 610823 | IF (1/1000) WB (1/1000) |
| Antibody | Mouse monoclonal anti-CD63 IgG1 | *Charrin et al., 2001* | TS63 | WB (1/1000) |
| Antibody | Mouse monoclonal anti-ADAM10 11G2 | *Arduise et al., 2008* | Diaclone: #857.800.000 | WB (1/1000) |
| Antibody | Rabbit polyclonal anti-alpha GAPDH | Sigma | #G9545 | WB (1/1000) |

*Appendix 1 Continued on next page*

*Appendix 1 Continued*

| Reagent type (species) or resource | Designation | Source or reference | Identifiers | Additional information |
|---|---|---|---|---|
| Antibody | Mouse monoclonal anti-actin, clone C4 | MP Biomedicals | SKU: 0869100-CF | WB (1/1000) |
| Antibody | Mouse monoclonal anti-alpha tubulin | Sigma | #T9026 | WB (1/2000) |
| Antibody | Rabbit polyclonal anti-ITGB1, ITGB4, ITGA4 | Cell Signaling | #4749 | WB (1/1000) |
| Antibody | Rabbit monoclonal anti-EGFR | Cell Signaling | #4267 | WB (1/1000) |
| Antibody | Rabbit polyclonal anti-Cx43 /GJA1 | Sigma | #C6219 | WB (1/3000) |
| Antibody | Mouse monoclonal anti-ANXA2 | Proteintech | #66035 | WB (1/2000) |
| Antibody | Mouse monoclonal anti-Alix, clone 3A9 | Biorad | #MCA2493 | WB (1/1000) |
| Antibody | Rabbit polyclonal anti-calnexin | Enzo | #SPA-860 | (WB 1/1000) |
| Sequence-based reagent | siRNA: non-targeting control | Origene | #SR30004 | |
| Sequence-based reagent | small double stranded RNA oligonucleotides targeting human CD9 | *Silvie et al., 2006* | | GAG CAT CTT CGA GCA AGA A- |
| Sequence-based reagent | small double stranded RNA oligonucleotides targeting human CD81 | *Silvie et al., 2006* | | CAC GTC GCC TTC AAC TGT A- |
| Commercial assay or kit | Lipofectamine RNAiMAX | Invitrogen | | Transfection of siRNAs |
| Commercial assay or kit | Lipofectamine 2000 | Invitrogen | | Transfection of plasmids in SH-SY5Y cells |
| Commercial assay or kit | Fugene HD | Promega | #E2311 | Transfection of plasmids in HEK293T and U2OS cells |
| Commercial assay or kit | Lenti-X Concentrator | TakaraBio | #631232 | |
| Commercial assay or kit | VivaSpin 20 | Cytiva | #28932360 | MWCO 10 kD |
| Software, algorithm | ProteoMap | https://www.proteomaps.net/; *Liebermeister et al., 2014* | | |
| Software, algorithm | DAVID | https://david.ncifcrf.gov/; *Sherman et al., 2022* | | |
| Software, algorithm | STRING | https://string-db.org/; *Szklarczyk et al., 2021* | RRID:SCR_005223 | |
| Software, algorithm | Cytoscape | https://cytoscape.org/ | RRID:SCR_003032 | |
| Software, algorithm | ICY | https://icy.bioimageanalysis.org/ | RRID:SCR_010587 | |
| Software, algorithm | PHOTOSHOP v23.5.5 | Adobe Systems | RRID:SCR_014199 | EVP diameter quantification |
| Software, algorithm | R package imp4p | https://rdrr.io/cran/imp4p/man/imp4p-package.html; *Giai Gianetto, 2021* | | |
| Other | DAPI stain | Invitrogen | D1306 | (1 µg/mL) |
| Other | WGA, Alexa Fluor-conjugated | Invitrogen | | (1/400) |

