## [Editor Report · eLife assessment]

Notario Manzano et al. offer a **valuable** first analysis of proteins within tunneling nanotubes (TNTs), membranous bridges connecting cells. This work distinguishes TNTs from extracellular vesicles, but further experimental and analytical tools are needed to refine the TNT proteome. **Solid** data supports a role for tetraspanins CD9 and CD81 in TNT function. The proposed model for CD9 and CD81 is over-interpreted and requires additional evidence for stronger support.

---

## [Referee Report · Reviewer #1 (Public Review)]

Summary:

The authors' claims that CD9 and CD81 are key regulators of TNT formation and function are well-supported by the data. The use of KO and OE models provides strong evidence. The differential proteomic analysis between TNTs and EVPs and the functional assays justify the conclusion that these tetraspanins are critical for TNT biogenesis and functionality. Overall, the manuscript presents a nice study that advances our understanding of TNTs and their regulation by CD9 and CD81. Despite some limitations, the strengths of the experimental design and the robustness of the data justify the authors' conclusions. Future studies addressing the identified weaknesses would further solidify these findings and their implications in pathological contexts.

Strengths:

Novelty and Significance - this study addresses the composition and regulation of tunneling nanotubes (TNTs). By identifying the roles of CD9 and CD81 tetraspanins, the researchers offer insights into the molecular mechanisms underlying TNT formation. This could have implications for understanding cellular communication in pathological conditions such as cancer.

Methodological Accuracy - the authors employed a well-designed biochemical approach to isolate TNTs from U2OS cells, distinguishing them from extracellular vesicles and particles (EVPs). The use of multiple independent preparations and the application of LC-MS/MS for proteomic analysis ensure robustness and reproducibility of the data.

Complete Analysis - the study provides a detailed proteomic profile of TNTs, identifying 1177 proteins and highlighting key components. The comparative analysis between TNTs and EVPs further strengthens the findings by demonstrating distinct proteomic landscapes.

Functional Insights - using knockout (KO) and overexpression (OE) models, the authors convincingly demonstrate the distinct roles of CD9 and CD81 in TNT formation and function. CD9 is shown to stabilize TNTs, while CD81 facilitates vesicle transfer, likely by aiding membrane docking or fusion.

Experimental Design - the use of actin chromobody-GFP and various fluorescent markers enabled the authors to visualize TNTs and validate their isolation protocol. Additionally, the combination of electron microscopy, flow cytometry, and live-cell imaging provided convincing evidence for their claims.

Weaknesses:

Potential Contaminations - while the authors took steps to minimize contamination with other cellular structures, the presence of some nuclear proteins and the possible inclusion of small portions of cell bodies or ER in the TNT preparations cannot be entirely ruled out. This may affect the interpretation of some proteomic data.

Limited Cell Models - the experiments were conducted in U2OS and SH-SY5Y cells. While these are relevant models, in vivo validation of the findings would significantly enhance the impact and translational potential of the research.

Functional Mechanisms - although the study provides strong evidence for the roles of CD9 and CD81, the exact molecular mechanisms by which these tetraspanins regulate TNT formation and vesicle transfer remain partially speculative. Further biochemical and biophysical analyses would be necessary to elucidate these mechanisms in detail.

---

## [Referee Report · Reviewer #2 (Public Review)]

Tunneling nanotubes (TNT) are common cellular protrusions that allow the transfer of multiple types of cargo between mammalian cells. TNTs are fragile, and lack any known unique marker, making it challenging to isolate and study them. Therefore, the content of TNTs is mostly unknown, and there are only a handful of proteins known to play a role in TNT formation or function.

In this paper, the authors developed a new protocol to isolate TNT fragments from a culture of adherent mammalian cells in a way that is distinctive of extracellular vesicle and identify the proteins within the TNT (referred to as TNTome) by mass spectrometry. The authors provide an analysis of the results in comparison to the extracellular vesicle (EV) proteome, and validate a few examples, thus providing valuable data for the TNT field. However, there is a big overlap between TNTome and EV proteome.

The authors further focus on two proteins, CD9 and CD81, that are enriched in TNTs. Using cells that are knocked out (KO) or over-expressing (OE) these proteins, the authors study their role in TNT formation and function. The authors focus on two major parameters, which are the percent of cells connected by TNT, and the percent of acceptor cells containing fluorescently labeled transferred vesicles. The authors use various assays, which are properly controlled, to measure these parameters. Their analysis provides convincing evidence that CD9 plays a partial role in TNT formation or stabilization and CD81 plays a partial role in forming fully elongated/connected TNT.

However, the authors overstate the importance of these proteins, since their absence only partially affects TNT formation and function, similar to what is seen when knocking out most any other protein implicated in TNT formation. Even their best results show just a 50% reduction for TNT formation and 70% vesicle transfer (in the double KO). Thus, these are not "key" regulators as the title suggests - no more than many other factors, some of them identified by the authors in previous publications. The model presented in Figure 7D is thus misleading, as it states that CD9 KO has "No TNT" which is incorrect (only a slight decrease according to Figure 3C), and states that CD81 KO has "Non-functional TNT" whereas there is still 50% vesicle transfer in this mutant.

In addition, the authors use vesicle transfer as a measure of function, but this is just one type of cargo amongst many others: ions, proteins, RNA, various organelles, and pathogens like viruses and bacteria. Since the authors clearly cannot test every type of cargo, the authors should at least be more accurate in their statements regarding functionality and mention the possibility that other types of cargo transfer could be less or more affected by the KO or OE of these proteins.

It is not completely clear from the text why the authors decided to focus on CD9 and CD81, which are also found in EV, instead of focusing on TNT-unique proteins, and in particular the cytoskeleton-related ones.

In summary, it is a good paper, that provides valuable data on the composition of TNT, and the role of additional players, bringing us closer to understanding the mechanism of TNT formation.

---

## [Referee Report · Reviewer #3 (Public Review)]

Initially, the authors isolated TNTs from EVPs and cell bodies of cultured U2OS cells. Using transmission electronic microscopy and nanoflow cytometry, they demonstrated that these two structures are morphologically different. In engineered cells, they observed the presence of actin and CD9 in TNTs by immunofluorescence. Then they employed mass spectrometry techniques to analyze the EVPs and TNT fractions, discovering that their compositions significantly differ and that CD9 and CD81 are abundant in both structures.

Subsequently, they studied the role of CD9 and CD81 in the formation of TNTs by using SH-SY5Y cells, first confirming their presence in TNTs via immunofluorescence. CD9 knockout (KO) cells, but not CD81 KO, exhibited a reduced percentage of cells connected via TNTs. The percentage of TNT-connected double KO cells was even lower compared to CD9 KO cells. Additionally, CD9 overexpression (OE), but not CD81 OE increased the percentage of TNT-connected cells.

The authors then investigated the influence of CD9 and CD81 on the capacity of cells to transport material through TNTs by quantifying vesicle delivery between cells. The percentage of acceptor cells containing vesicles (I call it here the efficiency of vesicle transfer) was reduced in CD9 KO cells and CD81 KO cells, and even lower in double KO cells. CD9 OE or CD81 OE increased vesicle transfer efficiency.

Then, they studied possible redundant or complementary roles in the formation of TNTs through a combination of KO and OE of CD9 and CD81 and observed that CD81 does not play any role in TNT formation when CD9 is present, and vesicle transfer of CD81 KO cells can be efficient in CD9 OE conditions.

Incubation of WT cells and CD81 KO cells with an anti-CD9 monoclonal antibody caused CD9 and CD81 clustering, significantly increasing the percentage of TNT-connected cells and duration of TNTs. While the antibody enhanced vesicle transfer efficiency in WT cells, it did not affect vesicle transfer in CD81 KO cells.

The article is well-written and addresses an important biological question, providing some insightful results. However, I have concerns regarding the connection between the experimental data and some of the conclusions drawn by the authors. Below I summarize my points:

- The protocol used to separate TNTs from EVPs and the cell body to determine their protein composition appears problematic. The authors apply mechanical stress by vigorously shaking the samples to achieve this separation. I am skeptical that this method robustly isolates TNTs from other cellular structures/components. I am concerned that their proteomic analysis might not be analyzing the composition of TNTs exclusively, but rather a mixture that includes other structures. For example, the second and eighth most abundant proteins identified are histones (Table S1), and about 20% of the total TNT proteins identified are either mitochondrial or nuclear proteins. The authors should attempt to improve the proteomics section of their study. To differentiate structural TNT proteins from debris, the authors could use statistical analysis to compare multiple independent preparations. Structural TNT proteins will likely be consistently present across all preparations, while non-structural TNT proteins may not. If this approach proves ineffective, the authors might need to refine their TNT isolation procedure.

- Throughout the whole manuscript, the authors quantify the percentage of cells connected by TNTs but do not provide data on the total number of TNTs, which would offer additional valuable information not captured by the percentage of TNT-connected cells alone.

- To study TNT functionality, the authors quantified the efficiency of vesicle transfer by calculating the percentage of acceptor cells containing donor vesicles. How was this percentage computed? The actual number of vesicles delivered to acceptor cells would provide a more accurate metric of vesicle transfer efficiency.

- In Figure 7D, the authors provide a working model. They claim that CD9 KO cells are incapable of forming TNTs. However, this is not supported by their data. The percentage of TNT-connected cells in CD9 KO cells is only slightly lower than in WT cells (Figure 3C).

- In the abstract and discussion of Figure 7D, the authors also claim that CD81 is necessary for the functional transfer of vesicles through TNTs by regulating membrane docking/fusion with the opposing cell. Furthermore, they propose in the discussion section that CD81 is involved in the opening of the TNT. However, all these claims are purely speculative and not supported by their data. If CD81 played such a role, vesicles would accumulate at the tip of the TNTs, which does not appear to be the case. Vesicle transfer occurs in CD81 KO cells. Additionally, TNT formation and efficient vesicle transfer are observed in CD81 KO cells and CD9 OE conditions, suggesting that docking/fusion is not dependent on CD81. Can the authors justify their claims? It is possible that CD81 KO cells might form TNTs with smaller diameters, potentially hindering vesicle transfer. Quantifying the dependence of TNT diameter on CD81 and CD9 expression would address this hypothesis.

- The authors should explain the implications of their study. They need to elaborate on how their findings could impact our understanding of cellular communication and potential applications in therapeutic strategies.

- Tetraspanins are involved in cell migration. In the CRISPR knockout experiments, could the observed changes in the percentage of TNT-connected cells be attributed to variations in cell migration potential?

- The reason behind the clustering of CD9 and CD81 after CD9 antibody treatment should be discussed.

---

## [Author Response]

We thank the reviewers for their careful reading of the manuscript and for their comments. Generally, we agree with the reviewers on the strengths and weaknesses of our manuscript. It is true that this work is a first step towards understanding the molecular mechanisms underlying TNT formation, and that further biochemical and biophysical analyses will be necessary to elucidate CD9 and CD81 roles. It also provides a toolbox for the future identification of important TNT factors, and perhaps biological markers.

However, we would like to better explain our choice of focusing on CD9 and CD81 in TNTs, given the fact that they are also expressed in EVPs. First, both were among the most abundant integral membrane proteins in TNTs, and overexpression of CD9 was previously shown to increase TNT number. However, a recent work directed by our coauthor E. Rubinstein clearly showed that the absence of CD9, CD81 or even both has minimal impact on the production or composition of EVs in MCF7 (Fan et al, Differential proteomics argues against a general role for CD9, CD81 or CD63 in the sorting of proteins into extracellular vesicles, J. Extracell Vesicles, 2023;12:12352. https://doi.org/10.1002/jev2.12352). This is in line with another recent publication (Tognoli, Commun biol 2023) and with our results showing that the concentration of EVPs was the same when CD9 was overexpressed, i.e. in conditions where TNT number and vesicle transfer were increased. Therefore, it is highly probable that the role of CD9 and CD81 in TNT vs. EVP formation is different, even if we cannot completely exclude a crosstalk between the two pathways.

Regarding the importance of CD9 and CD81 in TNT formation, our results are consistent with a non-exclusive regulation of the TNTs by these tetraspanins, and/or with partial compensatory mechanisms occurring in the absence of them by yet unknown factors. Interestingly, to our knowledge, none of the TNT regulators describedin the literature has a complete inhibitory effect when KO. These results confirm that several pathways can converge to regulate TNTs and are consistent with cellularplasticity. So it is hard to say whether factors like CD9 and CD81, which regulate TNTs and have other functions in cells, are “key” or simply “important”.

Finally, the model we present in Figure 7 is a schematic working model of possible CD9/CD81 roles, which is obviously simplified for ease of understanding. It is important to note that when we write “no TNT” above an empty space between 2 cells, this describes what is drawn, and corresponds to real conditions where fewer TNTs are detected. It was never our intention to over-interpret our data, but rather to make it clearer with this diagram, and we hope that reading the article will make this clear.